# Infection-derived lipids elicit an immune deficiency circuit in arthropods

Dana K. Shaw[1], Xiaowei Wang[1], Lindsey J. Brown[1,†], Adela S. Oliva Chávez[1], Kathryn E. Reif[2,†], Alexis A. Smith[3], Alison J. Scott[4], Erin E. McClure[1], Vishant M. Boradia[1], Holly L. Hammond[1], Eric J. Sundberg[5], Greg A. Snyder[5], Lei Liu[6], Kathleen DePonte[6], Margarita Villar[7], Massaro W. Ueti[2], José de la Fuente[7,8], Robert K. Ernst[1,4], Utpal Pal[3], Erol Fikrig[6,9] & Joao H.F. Pedra[1]

The insect immune deficiency (IMD) pathway resembles the tumour necrosis factor receptor network in mammals and senses diaminopimelic-type peptidoglycans present in Gram-negative bacteria. Whether unidentified chemical moieties activate the IMD signalling cascade remains unknown. Here, we show that infection-derived lipids 1-palmitoyl-2-oleoyl-*sn*-glycero-3-phosphoglycerol (POPG) and 1-palmitoyl-2-oleoyl diacylglycerol (PODAG) stimulate the IMD pathway of ticks. The tick IMD network protects against colonization by three distinct bacteria, that is the Lyme disease spirochete *Borrelia burgdorferi* and the rickettsial agents *Anaplasma phagocytophilum* and *A. marginale*. Cell signalling ensues in the absence of transmembrane peptidoglycan recognition proteins and the adaptor molecules Fas-associated protein with a death domain (FADD) and IMD. Conversely, biochemical interactions occur between x-linked inhibitor of apoptosis protein (XIAP), an E3 ubiquitin ligase, and the E2 conjugating enzyme Bendless. We propose the existence of two functionally distinct IMD networks, one in insects and another in ticks.

[1] Department of Microbiology and Immunology, University of Maryland School of Medicine, Baltimore, Maryland 21201, USA. [2] Animal Disease Research Unit, United States Department of Agriculture, Agriculture Research Service, Pullman, Washington 99164, USA. [3] Department of Veterinary Medicine, Virginia-Maryland Regional College of Veterinary Medicine, University of Maryland, College Park, Maryland 20742, USA. [4] Department of Microbial Pathogenesis, University of Maryland School of Dentistry, Baltimore, Maryland 21201, USA. [5] Institute of Human Virology, Departments of Medicine and Microbiology and Immunology, University of Maryland School of Medicine, Baltimore, Maryland 21201, USA. [6] Section of Infectious Diseases, Department of Internal Medicine, Yale University School of Medicine, New Haven, Connecticut 06510, USA. [7] SaBio. Instituto de Investigación en Recursos Cinegéticos IREC, CSIC-UCLM-JCCM, Ciudad Real 13005, Spain. [8] Department of Veterinary Pathobiology, Center for Veterinary Health Sciences, Oklahoma State University, Stillwater, Oklahoma 74078, USA. [9] Howard Hughes Medical Institute, Chevy Chase, Maryland 20815, USA. † Present addresses: U.S. Food and Drug Administration, White Oak Campus, Silver Spring, Maryland 20993, USA (L.J.B.); Center for Excellence in Vector Borne Diseases, Department of Diagnostic Medicine and Pathobiology, Kansas State University, Manhattan, Kansas 66506, USA (K.E.R.). Correspondence and requests for materials should be addressed to J.H.F.P. (email: jpedra@som.umaryland.edu).

The immune deficiency (IMD) signalling cascade is functionally analogous to the mammalian tumour necrosis factor (TNF) receptor network, and has a critical role in arthropod humoral responses[1,2]. In insects, peptidoglycan recognition protein LC (PGRP-LC) and PGRP-LE sense diaminopimelic-type peptidoglycans (DAP-PGN) present in most Gram-negative bacteria. PGRP-LC interacts with the protein IMD[3], which recruits the molecule Fas-Associated protein with Death Domain (FADD)[4]. FADD engages the caspase-8 homologue, death-related ced-3/Nedd2-like protein (DREDD)[5], which cleaves IMD and uncovers a binding site for lysine (K)63-polyubiquitylation via the E3 ubiquitin ligase Inhibitor of Apoptosis Protein (IAP)2[6]. Together with three E2 ubiquitin conjugating enzymes: Uev1a, Bendless (Ubc13) and Effete (Ubc5), IAP2 polyubiquitylates IMD in a K63-dependent manner. This leads to the recruitment of TGF-β activated kinase (TAK1) and Tak1-binding protein 2 (TAB2), which engage the I-κB kinase (IKK) complex[1,2]. The transcription factor Relish is then phosphorylated and the N-terminal portion (N-Rel) is cleaved by DREDD[2,6]. N-Rel is subsequently translocated to the nucleus and induces the production of antimicrobial peptides (AMPs)[2,6].

Variations of the IMD signalling cascade indicate the existence of an uncharacterized biochemical network. For instance, deficiency in components of the IMD pathway in *Drosophila* renders flies susceptible to Sindbis and Cricket paralysis viruses[7,8]. Unlike bacterial infections, activation of the IMD pathway by viruses does not result in a robust induction of AMPs[7,8]. Silencing the expression of *caspar*, an inhibitor of the IMD pathway, curbs parasite colonization by *Plasmodium falciparum* and *Leishmania* spp. in *Anopheles* mosquitoes and sand flies, respectively[9–12]. The genome of the Chagas disease arthropod vector, *Rhodnius prolixus*, does not encode IMD and FADD[13]. Nonetheless, targeted gene silencing of *relish* through RNA interference (RNAi) increased the population of the symbiotic bacterium *Rhodococcus rhodnii*[13]. Along these lines, genome sequencing of the pea aphid *Acyrthosiphon pisum* and the body louse *Pediculus humanus humanus* revealed gene losses in the IMD pathway[14,15] and, of particular importance to this study, comparative genomic analysis indicated the presence of an atypical IMD pathway in ticks[16–20].

These findings suggest the existence of plasticity in the IMD pathway of arthropods. Previously, we reported that the E3 ubiquitin ligase x-linked inhibitor of apoptosis protein (XIAP) restricts bacterial colonization of *Ixodes scapularis* ticks[21]. Herein, we used a combination of structural modelling, biochemical approaches and RNAi coupled to homology transfer and interactome analysis to demonstrate that XIAP interfaces with the IMD signalling pathway by interacting with the E2 conjugating enzyme Bendless. This molecular circuit functions despite lacking several upstream signalling components including the transmembrane PGRP receptor, the death domain protein FADD and the adaptor molecule IMD. The tick IMD pathway protects against colonization by two evolutionarily divergent bacteria: the Lyme disease spirochete *Borrelia burgdorferi* and the rickettsial pathogen *Anaplasma phagocytophilum*. Interestingly, infection-derived lipids 1-palmitoyl-2-oleoyl-*sn*-glycero-3-phosphoglycerol (POPG) and 1-palmitoyl-2-oleoyl diacylglycerol (PODAG) stimulate the *I. scapularis* IMD pathway. Moreover, immune priming with POPG and PODAG protect against infection by *A. phagocytophilum* and *A. marginale* in *I. scapularis* and *Dermacentor andersoni* ticks, respectively. These findings are conceptually important because they demonstrate that the immune system of ticks diverges from what has been demonstrated in insects.

## Results

**XIAP interfaces with the *I. scapularis* IMD pathway.** In an earlier study, we determined that the *I. scapularis* E3 ubiquitin ligase XIAP restricts colonization of the rickettsial bacterium *A. phagocytophilum* in ticks[21]. Here we optimized the purification of recombinant XIAP by testing a range of buffers for protein solubility (Supplementary Figs 1–2). We also validated XIAP structural integrity by circular dichroism (Supplementary Fig. 2d). The protein retained the previously characterized enzymatic activity, as assessed by polyubiquitylation assays. XIAP carried out K63-dependent polyubiquitylation (Supplementary Fig. 2e, lane 1), which was ablated when a point mutation at position 63 of ubiquitin was introduced (Ub$^{K63R}$; Supplementary Fig. 2e, lane 4). As expected, a point mutation at position 48 (Ub$^{K48R}$) had no effect on XIAP enzymatic activity (Supplementary Fig. 2e, lane 3). However, there was some residual polyubiquitylation in the absence of XIAP (Supplementary Fig. 2e, lane 7), which was attributable to the previously reported autocatalytic activity of the E2 conjugating enzyme UbcH13 (ref. 22).

To determine the signalling cascade in which XIAP interfaces, we performed structural modelling and compared our results with experimentally determined structures available in the protein data bank (PDB). We threaded XIAP onto the solved structure of the E3 ubiquitin ligase cellular inhibitor of apoptosis protein 1 (cIAP1; PDB: 3T6P; Supplementary Fig. 3a) and observed that XIAP carried a non-structured region and the catalytic Really Interesting New Gene (RING) domain, but did not have either the ubiquitin-associated (UBA) or the caspase activation and recruitment domain (CARD; Supplementary Fig. 3a). *I. scapularis* XIAP is substantially shorter than homologues found in humans, mice and *Drosophila* and has different domain distributions. The tick XIAP carries only one conserved baculoviral IAP repeat (BIR) and no annotated UBA domains (Supplementary Fig. 3b)[21]. The predicted structure of *I. scapularis* XIAP revealed a model of high quality with conserved residues in the BIR domain when compared with *Drosophila*, mice and humans (Supplementary Fig. 3c–d)[23].

The tick XIAP BIR domain carried the typically conserved zinc coordinating residues (Supplementary Fig. 3a in cyan and yellow and Supplementary Fig. 3d). Alignment of human and *Drosophila* BIR domains showed that the *I. scapularis* XIAP BIR domain belonged to the type III group (Supplementary Fig. 3e). The tick XIAP type III BIR domain demonstrated a preference for proline in the third residue of the ligand (Supplementary Fig. 3e), resembling the classic IAP-binding motif (Supplementary Fig. 3f)[23]. These findings provided the impetus to perform homology transfer between the tick XIAP, its homologue in humans, and the closely related protein melanoma (ML)-IAP. Homology transfer is the transposition of a function from one protein to another on the basis of their common evolutionary origin[24]. This method proved instrumental for functional prediction because there is a lack of empirically determined data. We acquired the top related proteins interacting with the human XIAP and ML-IAP based on previously observed protein and genetic interactions, pathways and co-localization assays (Fig. 1a,b; Supplementary Table 1). *I. scapularis* homologues showed an overrepresentation of immune-related genes in the XIAP interactome (Fig. 1c; Supplementary Fig. 4). In particular, six out of 14 proteins (~43%) [$P = 0.01$; GO:0002376 and GO:0006955] were identified from the IMD pathway: (1) Bendless, (2) Effete, (3) Uev1a, (4) IAP2, (5) TAK1 and (6) TAB2 (Fig. 1c). Overall, these findings suggested that XIAP interfaces with the IMD signalling pathway in *I. scapularis* during microbial infection.

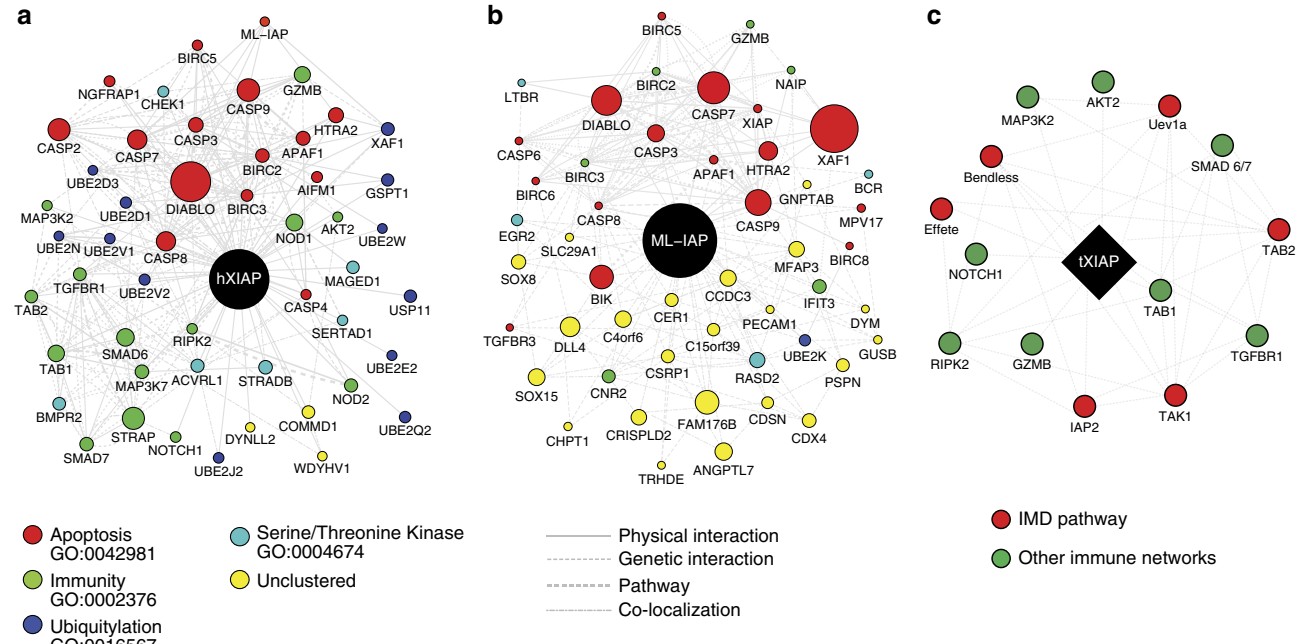

**Figure 1 | The *I. scapularis* XIAP interactome.** Top 50 related genes interacting with the human (**a**) XIAP and (**b**) ML-IAP were compiled in a network according to GeneMANIA. Interactomes were visualized by using Cytoscape. Networks and biological pathways were integrated based on previously observed protein and genetic interactions, pathway, and co-localization assays. Proteins were grouped according to the GO functional categories available at DAVID. Size corresponds to the score given to each node. Edge type is based on the interaction map. Width is determined by scored evidence of interaction. (**c**) *I. scapularis* homologues were obtained with PSI-BLAST and GeneCard searches based on the human XIAP or the ML-IAP interactome. Sixty-eight candidates were grouped according to the GO functional categories available at DAVID. Immunity genes are represented in green and all candidates highlighted in red are predicted to be involved in the arthropod IMD pathway. See also Supplementary Figs 1–4 and Supplementary Table 1.

**XIAP interacts with the E2 conjugating enzyme Bendless.** To validate the findings described above, we analysed the quantitative proteomics data deposited in the Dryad repository database[25]. We identified the IMD pathway E2 conjugating enzymes Bendless, Uev1a and Effete as being differentially expressed on *A. phagocytophilum* infection in the midgut of *I. scapularis* (Fig. 2a). We then used the recombinant protein XIAP (XIAP-GST) cross-linked to a glutathione agarose column to perform pull-down assays with tick cell lysates. Cross-linking did not cause steric hindrance or interfere with enzymatic activity (Supplementary Fig. 2f). Bound proteins were eluted and a peptide identified by tandem mass spectrometry resembled the E2 conjugating enzyme Bendless from the IMD pathway. Importantly, Bendless shares homology with UbcH13 ($E$ value $= 1e^{-101}$), the E2 ubiquitin conjugating enzyme used in our assays (Supplementary Fig. 2e,f). Furthermore, when we docked the tick protein Bendless to XIAP *in silico*, the resulting model indicated that these two molecules could interact (Fig. 2b).

To confirm these results, we first incubated the recombinant forms of tick Bendless and XIAP and analysed their interactions using mobility shift assays[26]. Under non-denaturing conditions, XIAP was shifted to a higher molecular weight with increasing concentrations of Bendless (Fig. 2c, Supplementary Fig. 8). This observation was substantiated with an ELISA-based approach to assess binding saturation of Bendless to XIAP. With XIAP as a bait protein, increasing concentrations of Bendless bound to XIAP reaching saturation at 9.1 µM (Fig. 2d). Furthermore, Bendless and XIAP interactions were blocked with a mouse monoclonal antibody to the human homologue of Bendless in a dose-dependent manner (Fig. 2e). Next, we observed that the recombinant forms of the tick XIAP and Bendless were able to produce free K63-, but not, K48-polyubiquitin chains in an enzymatic reaction. These results were displayed with the use of

wildtype ubiquitin (Ub^WT; Fig. 2f, lanes 1–3 and 5–10, Supplementary Fig. 9a,b); or, alternatively, with the addition of ubiquitin containing mutations at either lysine 63 (Ub^K63R; Fig. 2f, lane 11, Supplementary Fig. 9c) or lysine 48 (Ub^K48R; Fig. 2f, lane 12, Supplementary Fig. 9c).

We then employed a Human Embryonic Kidney (HEK)293 T cell transfection system with plasmids expressing the tick XIAP and Bendless (Fig. 2g, Supplementary Fig. 10a–c). Immunoblotting against FLAG and HA tags (Bendless-FLAG and XIAP-HA) demonstrated robust protein expression for Bendless and XIAP in transfected cells. When co-expressed, immunoprecipitation against the affinity tags revealed that Bendless specifically pulled down XIAP and vice versa (Fig. 2g, Supplementary Fig. 10d–e). Finally, to assess whether XIAP-Bendless interactions could occur *in vivo*, we extracted protein from unfed *I. scapularis* nymphs that had been microinjected with either siRNA targeting *bendless* (siBendless) or a scrambled control (scBendless). Whole tick lysates were used as bait and were incubated with increasing concentrations of recombinant XIAP. We observed that protein extracted from control ticks (scBendless) had significantly higher amounts of bound XIAP when compared with tick lysates silenced with *bendless* (scBendless) (Fig. 2h). Altogether, we demonstrated that XIAP and Bendless directly and specifically interact with each other through six independent approaches.

**The IMD pathway restricts bacterial colonization in ticks.** XIAP restricts *A. phagocytophilum* colonization of *I. scapularis* and, when silenced, confers a survival advantage for this rickettsial bacterium (Fig. 3a)[21]. Because Uev1a activates the human homologue of Bendless, UbcH13, to carry out polyubiquitylation[27,28], we hypothesized that the same would be true for ticks. We, therefore, employed a dual knock down

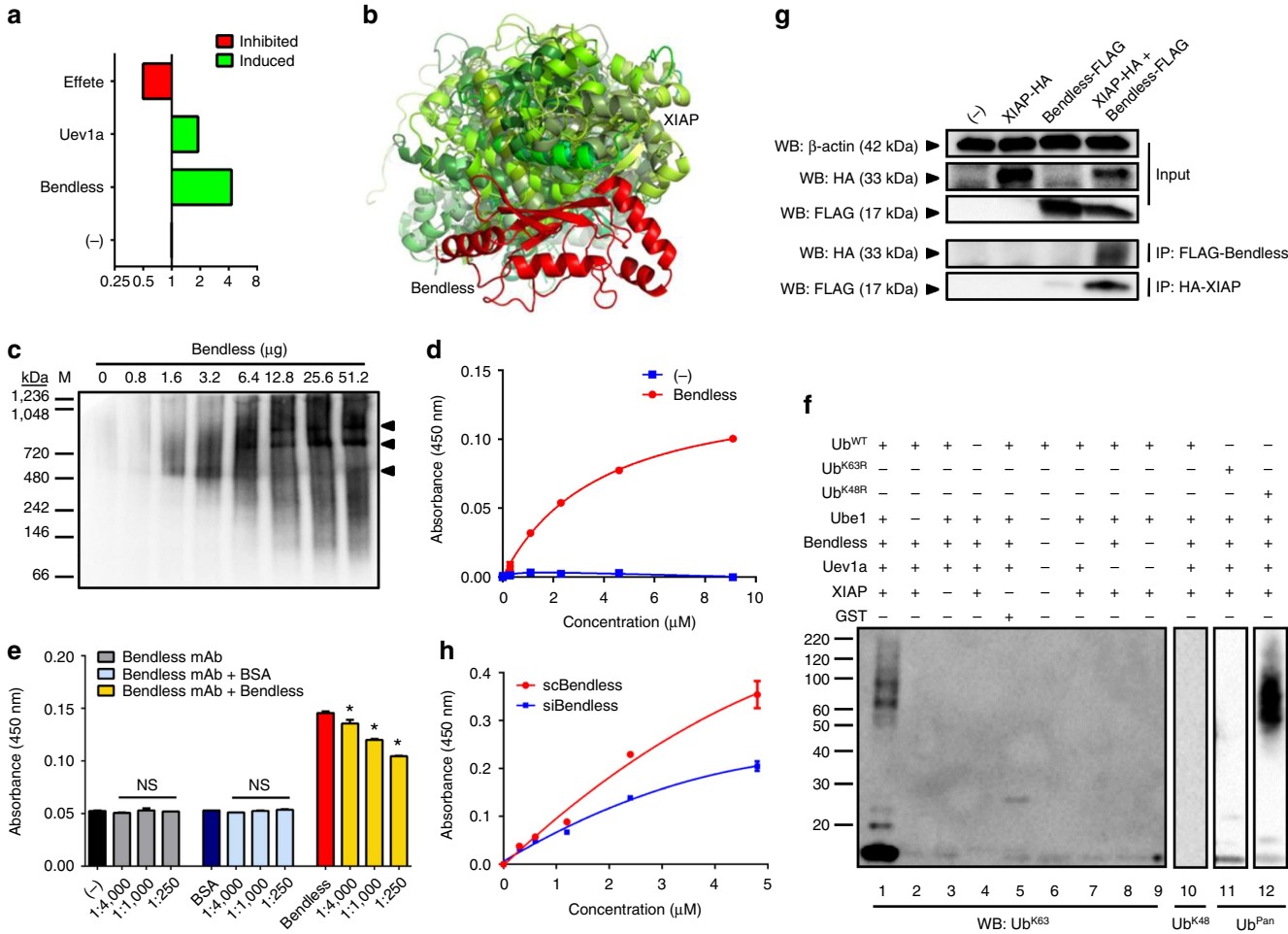

**Figure 2 | Bendless-XIAP molecular interactions.** (**a**) Protein differential expression on *A. phagocytophilum* infection of *I. scapularis* ticks based on iTRAQ proteomics data deposited on the Dryad repository database. (**b**) Structural docking model demonstrating the interaction between *I. scapularis* XIAP and the UbcH13 homologue, Bendless. (**c,d**) Native gel and ELISA analysis of a fixed amount of recombinant (0.2 μg) XIAP incubated with increasing amounts of recombinant Bendless. The analysis shown is one of two biological replicates. (**e**) XIAP-Bendless binding inhibition with a monoclonal antibody against the human homologue of Bendless (UbcH13). (**d,e**) The average of two technical replicates are plotted. (**f**) XIAP polyubiquitylation assay with recombinant Bendless (lane 1). Control conditions were performed in the absence of an E1 enzyme—Ube1 (lanes 2 and 6), XIAP (lanes 3, 5 and 6), wild-type ubiquitin (lanes 4, 11 and 12), Bendless (lanes 6, 7 and 9) and Uev1a (lanes 6, 8 and 9). Immunoblots were probed with antibodies specific for K63- (lanes 1–9) and K48- (lane 10) polyubiquitin chains or with a pan-ubiquitin antibody (lanes 11 and 12). GST was used as a negative control. The Western blot (WB) shown is one of two biological replicates. (**g**) Immunoprecipitation (IP) analysis followed by WB showing the interaction between XIAP and Bendless within HEK293 T cells transfected with the indicated vectors. Input indicates normalizing amounts. The WB shown is one of two biological replicates. (**h**) ELISA with protein extracts from unfed *I. scapularis* nymphs that were microinjected with either a scrambled control (scBendless) or siRNA specific for *bendless* (siBendless) and incubated with increasing amounts of recombinant XIAP. Points are the average of 10 biological replicates with two technical replicates each. See also Supplementary Figs 1–3 and 8–10.

scheme targeting both *uev1a* and *bendless* to assess the contribution of these enzymes in the *I. scapularis* IMD pathway during *A. phagocytophilum* infection. Significant silencing was achieved for both *bendless* and *uev1a*, which caused increased *A. phagocytophilum* burden in tick cells when compared with the control treatment (Fig. 3b). Silencing the positive regulator of the IMD pathway, *relish*, also favoured *A. phagocytophilum* infection of tick cells (Fig. 3c), whereas the converse results were obtained when we knocked down the expression of *caspar*, a negative regulator of the IMD pathway (Fig. 3d). Reduced *caspar* expression should cause the pathway to be over-activated and, accordingly, we observed decreased *A. phagocytophilum* colonization of tick cells when *caspar* was silenced (Fig. 3d).

To determine whether the results obtained *in vitro* could also be observed *in vivo*, we placed *I. scapularis* ticks microinjected with siRNA for *bendless/uev1a*, *relish*, *caspar* and scrambled

controls on mice and allowed them feed to repletion (Fig. 4). We determined gene silencing and *A. phagocytophilum* load as a function of 16 s rDNA in fully-engorged *I. scapularis* nymphs. As observed for the ISE6 cell experiments, *I. scapularis* ticks microinjected with the siRNA for *bendless/uev1a* and *relish* were more susceptible to *A. phagocytophilum* infection when compared with the control treatment (Fig. 4a,b). Conversely, silencing *caspar* reduced *A. phagocytophilum* infection of ticks (Fig. 4c). To ascertain whether the IMD pathway of ticks responded to additional Gram-negative pathogens, we infected *I. scapularis* ticks with *B. burgdorferi*, the causative agent of Lyme disease. RNAi silencing of *bendless/uev1a*, *relish* and *caspar* altered *B. burgdorferi* colonization of *I. scapularis* in a manner similar to *A. phagocytophilum* (Fig. 4d–f). In sum, we discovered that the atypical IMD signalling pathway restricts *A. phagocytophilum* and *B. burgdorferi* colonization of *I. scapularis* ticks.

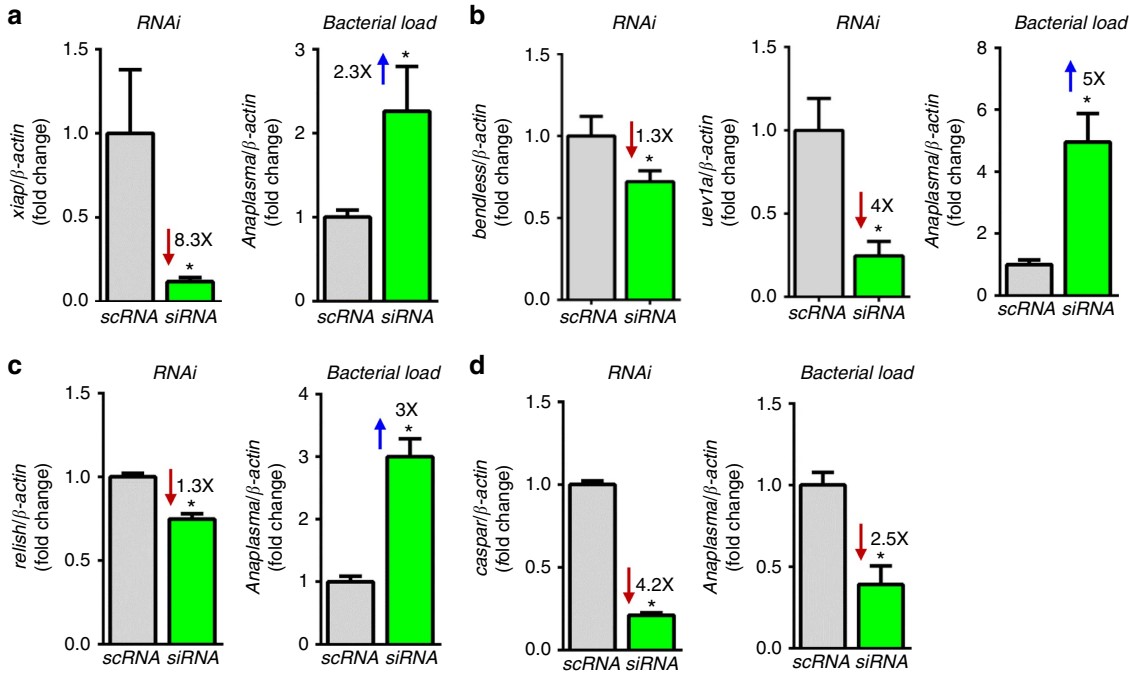

**Figure 3 | The *I. scapularis* IMD pathway responds to *A. phagocytophilum* infection.** (**a**–**d**) ISE6 ($1 \times 10^5$) cells were infected with *A. phagocytophilum* (MOI 50) following targeted gene silencing of (**a**) *xiap*, (**b**) *bendless/uev1a*, (**c**) *relish* and (**d**) *caspar*. Gene silencing and *A. phagocytophilum* load (16 s rDNA) were measured by quantitative reverse transcriptase–PCR at 18 h post-infection in ISE6 cells. Replicates of 5 were expressed as means ± s.e.m. All experiments shown are representative of five biological replicates with two technical replicates each. Student's t test. *$P < 0.05$. scRNA, scrambled RNA; siRNA, small interference RNA. See also Supplementary Table 2.

**Infection-derived lipids stimulate the IMD pathway**. Both *B. burgdorferi* and *A. phagocytophilum* induced expression of AMPs triggered by the IMD but not the Toll pathway in *Drosophila* (Supplementary Fig. 5). These results were intriguing because neither *A. phagocytophilum*[29] nor *B. burgdorferi*[30] have DAP-PGN in the cell envelope, which is the canonical agonist of the IMD pathway[1,2]. They instead use lipids and lipid-containing molecules for structural support of the membranes[31,32]. We sought to determine the unidentified chemical moiety that activates the IMD signalling cascade in these systems. Hence, we conducted an unbiased lipid analysis using matrix-assisted laser desorption/ionization (MALDI)-time of flight (TOF) mass spectrometry of host-free *A. phagocytophilum*, as we hypothesized that lipids could be activating the tick IMD pathway (Supplementary Table 3). A glycerophospholipid putatively identified as phosphatidylglycerol - PG 34:1 (34 total acyl carbons and one unsaturation) was enriched in *A. phagocytophilum*-infected samples when compared with host cells alone (Supplementary Table 3). We used a pure standard of 1-palmitoyl-2-oleoyl-*sn*-glycero-3-phosphoglycerol (POPG; also PG 34:1) for stimulation studies (Fig. 5a). Two control lipids were also selected: 1-palmitoyl-2-oleoyl diacylglycerol to match the acyl arrangement of POPG, but lacking a headgroup (PODAG; DG 34:1), and 1-myristoyl-2-palmitoyl-*sn*-glycero-3-phosphocholine (MPPC; PC 30:0) to serve as a negative control with an unmatched acyl arrangement and an unrelated headgroup (Fig. 5a).

To assess whether POPG, PODAG and MPPC could stimulate humoral immune pathways, we used the *Drosophila* surrogate model because pathway-specific AMPs have not yet been identified in *I. scapularis*. We stimulated *Drosophila* cells with increasing concentrations of lipids and assessed activation of either the IMD or Toll pathways by quantifying transcripts of specific AMPs: *diptericin* (IMD) or *im1* (Toll). None of the three

lipids affected the Toll pathway (Fig. 5b). However, two out of the three lipids, POPG and PODAG, caused a dose-dependent increase in *diptericin,* while MPPC-stimulated cells remained unchanged (Fig. 5c). Altogether, these findings indicate that POPG and PODAG specifically stimulate the IMD pathway.

To evaluate whether this stimulatory effect on the IMD pathway also occurred in ticks, we used an approach that involved first priming with the lipids and then infecting cells with *A. phagocytophilum*. We rationalized that pre-stimulation of tick cells with POPG and PODAG would induce the activation of the IMD pathway, conferring a survival disadvantage for the bacteria. Accordingly, *Drosophila* and ISE6 cells were primed with 1 ng of each lipid for six hours, corresponding to the characterized peak of IMD pathway activation[33]. Cells were then infected with *A. phagocytophilum* overnight. In agreement with previous experiments, cells treated with POPG led to a statistically significant reduction in bacterial load when compared with the naive group both in *Drosophila* and *I. scapularis* (Fig. 5d,e). Conversely, MPPC stimulation did not affect bacterial survival inside of cells when compared with the control treatment (Fig. 5d,e). PODAG, on the other hand, only conferred a protective effect in tick cells. Although we do not know the biological significance of these findings, we speculate that, as there is a divergence in the IMD signalling pathway across arthropods[34], it is possible that ticks and insects respond to pathogen associated molecular patterns (PAMPs) differently on microbial infection.

We raised an antibody against the positive IMD pathway regulator, Relish. Cleavage of Relish by DREDD can be used as a rapid read-out for the activation of the IMD pathway[2,6]. In the tick system, Relish cleavage occurred very rapidly after DAP-PGN stimulation (Fig. 5f, Supplementary Fig. 11). Similarly, both *A. phagocytophilum* and infection-derived lipids, POPG and PODAG, induced the appearance of Rel-N as early as one-minute

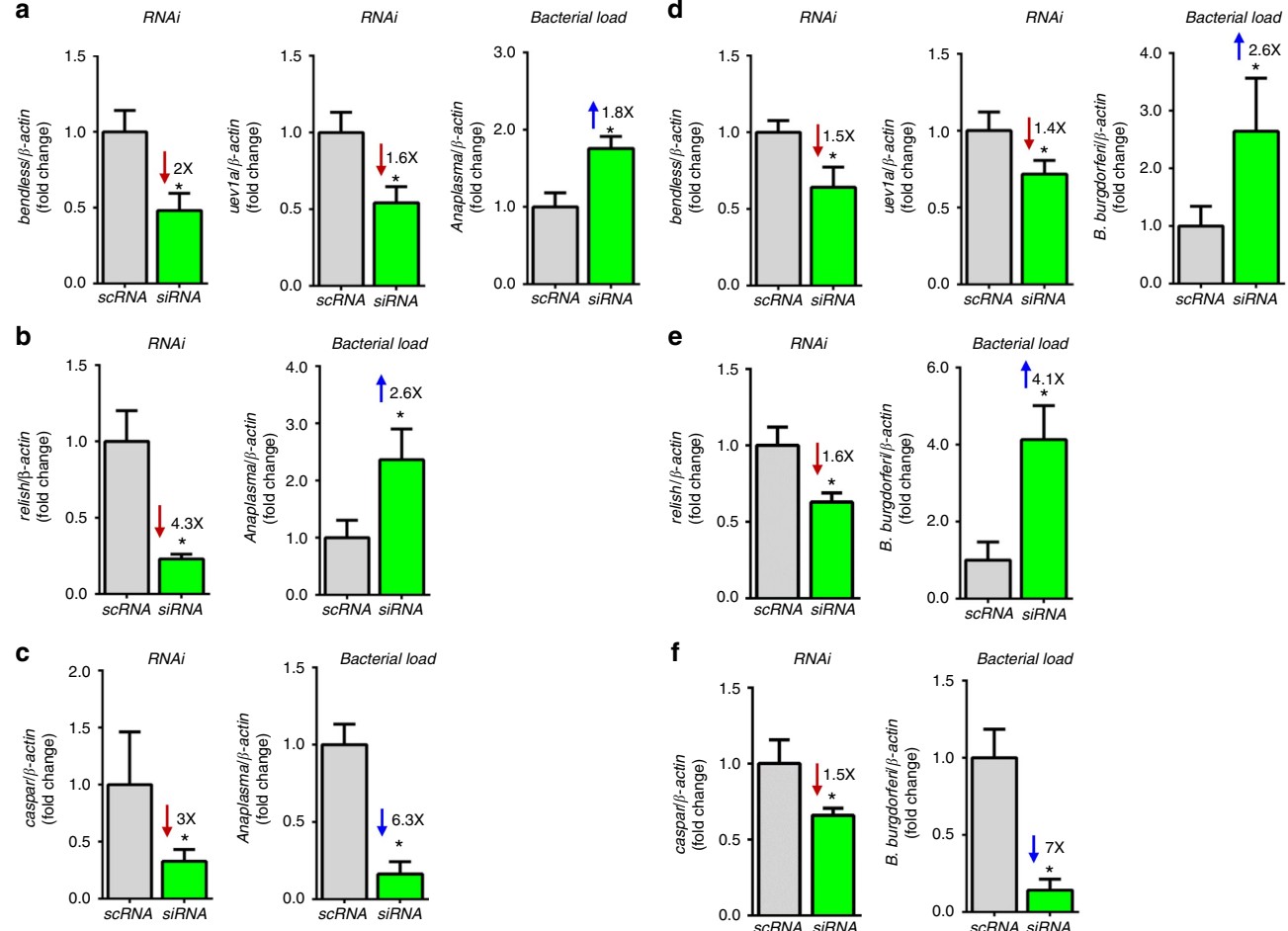

**Figure 4 | The *I. scapularis* IMD pathway affects bacterial colonization *in vivo*.** (**a**) RNAi silencing of *bendless/uev1a, relish* and *caspar* in *I. scapularis* nymphs following tick feeding on (**a–c**) *A. phagocytophilum*-infected or (**d–f**) *B. burgdorferi*-infected mice. Silencing levels and bacterial load were measured six days post-infection in whole *I. scapularis* nymphs. Samples represent the mean of 5–10 individual ticks, two technical replicates each, ± standard errors of the means (SEM). Student's t test. *P < 0.05. scRNA, scrambled RNA; siRNA, small interference RNA. See also Supplementary Table 2.

post-stimulation of the *I. scapularis* tick cell line ISE6 (Fig. 5f, Supplementary Fig. 11). This effect was also dose dependent (Fig. 5g, Supplementary Fig. 12). Altogether, these data suggested a molecular mechanism linking *A. phagocytophilum* infection with the activation of the of *I. scapularis* IMD pathway.

The priming with POPG and PODAG in ticks was not due to off-target signalling mechanisms. Knock-down of molecular components of the tick IMD, but not the Toll or the JAK-STAT pathways, abolished the effect that lipid priming had on bacterial survival inside tick cells (Fig. 6a–h). Silencing the expression of the heterodimeric E2 ubiquitin conjugating complex, *bendless/uev1a* or the E3 ubiquitin ligase *xiap* hampered protection against *A. phagocytophilum* infection (Fig. 6a–d). Conversely, targeted-gene silencing of the Toll and JAK-STAT pathways (siRNA) had no altered phenotype when compared with the control group (scRNA) (Fig. 6e–h). Collectively, our results demonstrate a mechanism by which two lipid agonists (POPG and PODAG) stimulate the IMD pathway of *I. scapularis* ticks.

We examined whether lipid priming offered bacterial cross-protection in another chelicerate model system. We inoculated a calf with the most prevalent tick-borne livestock pathogen *A. marginale* and allowed mock or *Dermacentor andersoni* ticks injected with POPG, PODAG and MPPC to feed. After feeding, ticks were removed and *A. marginale* load was measured six days post-feeding. As previously observed for the

*A. phagocytophilum-I. scapularis* system, POPG and PODAG but not MPPC priming, was protective against bacterial infection of ticks (Fig. 6i). These results suggested that the atypical IMD signalling pathway was also functional in ticks of veterinary importance.

**PGRP knockdown does not affect *A. phagocytophilum* infection.** In insects, PGRPs can function as immune pathway receptors, negative regulators of the immune response or as effectors that kill bacteria by enzymatically breaking down peptidoglycans[34]. As previously mentioned, the tick genome does not encode a transmembrane PGRP-LC, which is the known IMD pathway receptor in insects. However, there are four PGRPs that are predicted to be either extracellular with amidase activity (PGRP-4: XM_002413046.1) or intracellular and non-catalytic (PGRP-1: XM_002411731.1, PGRP-2: XM_002433644.1, PGRP-3: XM_002410377.1)[34]. To investigate whether the encoded tick PGRPs interfaced with the IMD pathway during *A. phagocytophilum* infection, we silenced each PGRP individually or in combination to address potential redundancy (Supplementary Fig. 6). Although significant silencing was achieved, no difference in bacterial load was observed for any of the treatments (Supplementary Fig. 6a–e). This was in agreement with the PGRP literature, which describes affinity for peptidoglycan, but no known lipid-binding capabilities[2,34].

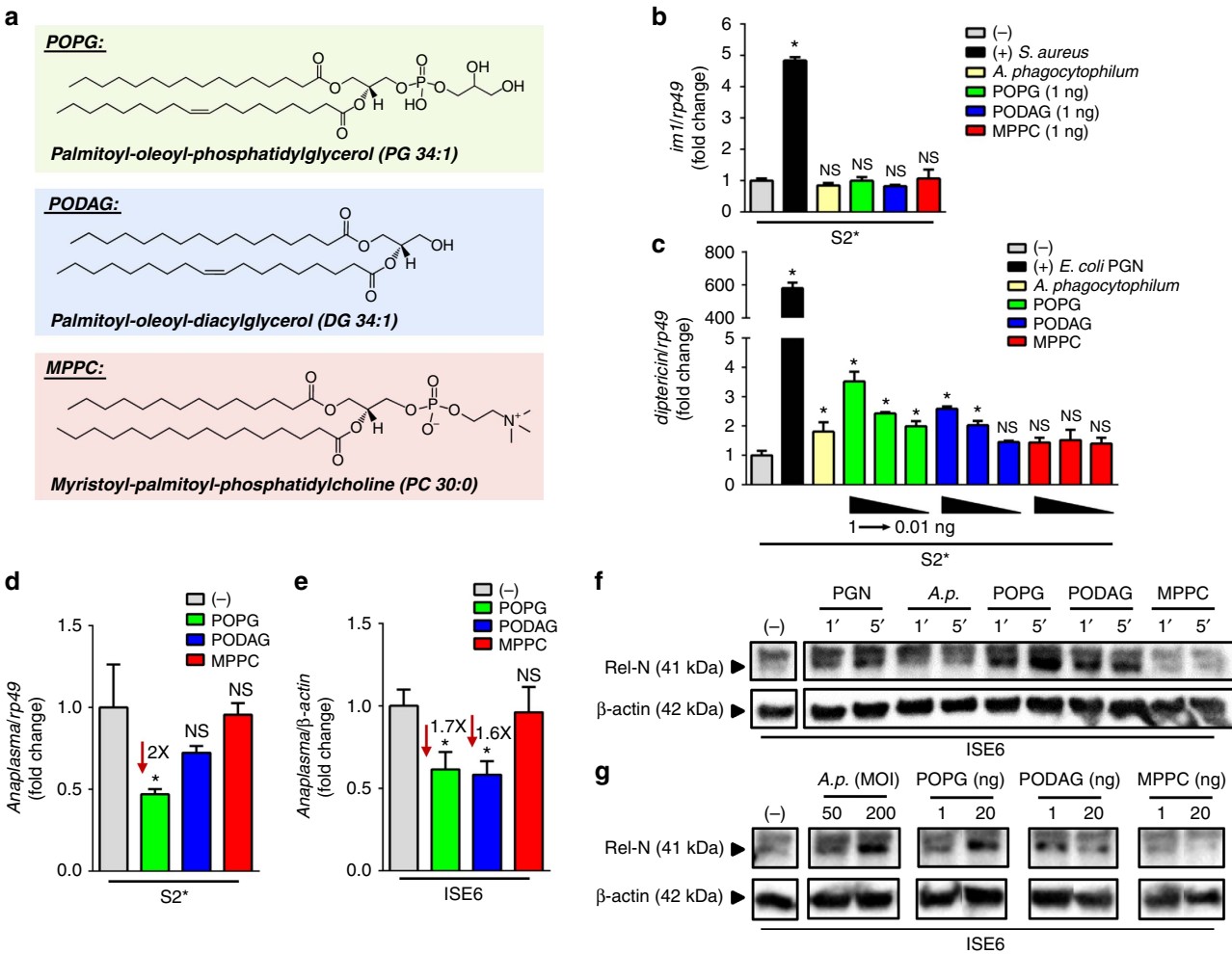

**Figure 5 | Infection-derived lipids stimulate the IMD pathway.** (**a**) Reference structures for the three lipids used in stimulation studies: (1) POPG, (2) PODAG and (3) MPPC. (**b,c**) Triplicate samples of $1 \times 10^6$ S2* cells were primed with 20-hydroxyecdysone (1 μM) and stimulated with 0.01–1 ng of indicated lipids, *A. phagocytophilum* (MOI 50) and positive controls for the Toll pathway (*S. aureus*) and the IMD pathway (*E. coli* peptidoglycan). Quantitative PCR (qPCR) quantifying *diptericin* and *im1* transcripts are shown. (**d**) Triplicate samples of $1 \times 10^6$ S2* cells or (**e**) Five replicates of $1 \times 10^5$ ISE6 cells were incubated with 1 ng of indicated lipids before *A. phagocytophilum* infection at MOI 50. Bacterial load was quantified by qPCR and normalized to either *rp49* (*Drosophila*) or *β-actin* (ISE6 tick cells). Data are represented as the mean ± s.e.m. Analysis of variance-Dunnet. *$P < 0.05$. NS, not significant. ( − ), non-stimulated. ISE6 ($1 \times 10^6$) cells were stimulated with (**f**) diaminopimelic-type peptidoglycan (PGN) (10 μg/ml), *A. phagocytophilum* (MOI 50) and indicated lipids (1 ng) at indicated times or (**g**) the indicated ranges of *A. phagocytophilum* and lipids for 15 min. Lysates were probed against an *I. scapularis* Relish polyclonal antibody (Rel-N 41 kDa). β-Actin (45 kDa) was used as a loading control. (**b–e**) Data are representative of 3–5 biological replicates, as indicated, and two technical replicates. (**f–g**) Western blots (WBs) shown represent one of three biological replicates. See also Supplementary Figs 5, 11 and 12 and Supplementary Tables 2 and 3.

**IMD pathway divergence among arthropod subphylums.** *I. scapularis* does not bear some components of the IMD pathway, such as transmembrane PGRPs and the signalling molecules IMD and FADD (Supplementary Fig. 7)[16–20,34]. Of significant interest, these observations are not specific to ticks, which became evident as we mined other arthropod genomes and a clear immunological pattern emerged between the branches of Arthropoda. While *imd* was mostly present in the Pancrustacea (Hexapods and Crustaceans), this gene was absent in Myriapods (centipedes and millipedes) and Chelicerates (ticks, spiders, mites and scorpions) (Fig. 7a). This phylogenetic relationship was consistently seen in other analyses such as with Relish, the transcription factor of the IMD pathway (Fig. 7b), and with the PGRPs (Fig. 7c), reflecting a clear divergence between Pancrustacea and Chelicerates/Myriapods. Altogether, our results provided strong evidence that two functionally distinct IMD networks exist: one previously recognized by the scientific

community occurring in Hexapods and Crustacea, and another atypical pathway displayed in Chelicerates and Myriapods (Supplementary Fig. 7).

## Discussion

The prevailing view of humoral immunity in arthropods is largely driven by studies performed in Dipteran insects[2,16,18–20]. The assumption is that pattern recognition receptors sense pathogens and/or danger signals, which then trigger an immune response similar to what has been described in model organisms[16,19,20,34]. Although this paradigm has certainly advanced our knowledge of arthropod immunity, this premise carries limitations when pathways in evolutionarily distant species, such as ticks, do not resemble what has been described for insects. For example, ticks do not carry βGRPs[16], which, in *Drosophila*, binds to the polysaccharide β-1,3 glucan from the cell wall of fungi and the

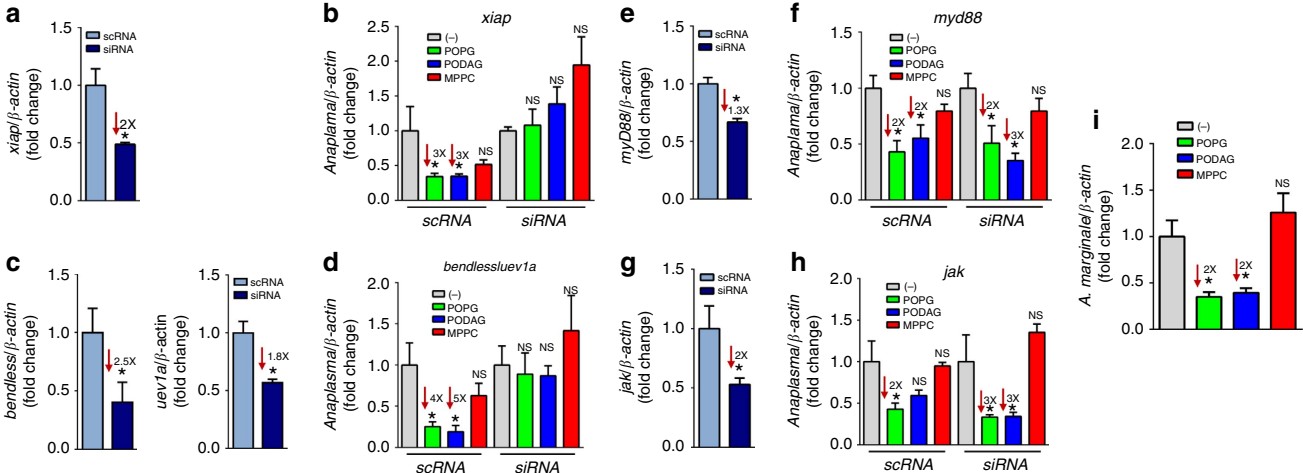

**Figure 6 | Lipid priming is protective against bacterial colonization of ticks. (a–h)** Five replicates of $1 \times 10^5$ ISE6 cells were transfected with siRNA molecules targeting components of the *I. scapularis* immune system. **(a,c,e,g)** Silencing efficiency or **(b,d,f,h)** *A. phagocytophilum* load was measured for the components of the **(a–d)** IMD, **(e–f)** Toll and **(g–h)** JAK-STAT pathways. Transfected cells were incubated with 1 ng of indicated lipids for 6 h and infected with *A. phagocytophilum* at MOI 50. Bacterial burden was quantified and normalized against β-actin. Data are represented as the mean ± standard errors of the means (SEM). Analysis of variance (ANOVA)-Dunnet; Student's *t*-test. *$P < 0.05$. NS, not significant. ( − ), non-stimulated. Data are representative of 5 biological replicates and two technical replicates. **(i)** *D. andersoni* ticks were mock- or lipid-injected (1 ng). Ticks were allowed to feed in individual group patches on a splenectomized, acute, *A. marginale*-infected calf for six days. Midguts from individual ticks were assessed for *A. marginale* infection levels by quantitative reverse transcriptase–PCR. Bacterial burden was quantified and normalized against β-*actin*. Samples represent the mean of 15-20 individual ticks ± s.e.m. ANOVA-Dunnet. *$P < 0.05$. NS, not significant. ( − ), non-primed. See also Supplementary Fig. 6 and Supplementary Tables 2 and 3.

lipopolysaccharide of Gram-positive bacteria[1,2]. Moreover, *Ixodidae* ticks do not have a pro-phenoloxidase system, which is essential for pathogen control in insects by melanization[17,34–36].

In this article, we have demonstrated an immune signalling cascade in ticks with several conserved molecules from the insect IMD pathway (Bendless/Uev1a, Relish and Caspar). However, the tick IMD network also lacks upstream signalling components, such as the PGRP-LC receptor and the signalling molecules, FADD and IMD[17,34]. This pathway responds to infection-derived lipids, POPG and PODAG, and does not involve the encoded PGPRs in the *I. scapularis* genome during *A. phagocytophilum* infection. Interestingly, although *I. scapularis* does not encode a transmembrane PGRP-LC receptor, Relish is still cleaved in response to DAP-PGN exposure. It has been previously shown that soluble PGRPs are capable of recognizing DAP-PGN and may assist in activating the *Drosophila* IMD pathway by providing a co-receptor function to transmembrane PGRPs[3]. Because tick cells have only soluble PGRPs encoded in the annotated genome[17,34], they may be able to recognize and respond to DAP-PGN, particularly, if invaded by intracellular bacteria.

Our findings suggest that the immune system of Chelicerates and Myriapods is fundamentally different when compared with Hexapods and Crustaceans (Fig. 7 and Supplementary Fig. 7). The conceptual implications of these results are wide in scope because it suggests that atypical IMD signalling cascades exist across Arthropoda. The notion that immune pathways in ticks diverge from insects may be expected, given their unique lifestyle when compared with other blood-feeding arthropods. For example, ticks are obligate hematophageous parasites, feeding exclusively on blood at all life stages, and are capable of transmitting a variety of disease-causing agents, including bacteria, viruses and protozoa[37,38]. The diverse pathosphere and relatively long life span of ticks, which can extend over 10 years for some species[35], suggests that unique evolutionary pressures exist for the development of immune signalling networks to control pathogens and promote prolonged survival.

Ticks are one of the earliest lineages of terrestrial arachnids, estimated to have originated between 443 and 120 million years ago[35,39–41]. Owing to their ancient evolutionary history, there is potential for the use of ticks as model organisms to study fundamental questions in arthropod immunology as well as in higher organisms. One can envision a scientific approach where conceptual breakthroughs made in ticks can be applied to other organisms. This possibility is supported by our observation that POPG, PODAG and organisms without DAP-PGN (*A. phagocytophilum* and *B. burgdorferi*) stimulate upregulation of the *Drosophila* IMD pathway-specific AMP *diptericin*, suggesting a conserved IMD pathway across arthropods. This combinatorial strategy will: (i) permit the identification of host and microbial factors that induce or suppress immune signalling; (ii) lay the groundwork for novel insights in pathogen-vector interactions; and (iii) help to develop novel interventions for prevention of tick-borne diseases.

## Methods

**Bacteria and animal models.** *I. scapularis* nymphs were obtained from the Bio-defense and Emerging Infectious Diseases (BEI) Research Resources Repository from the National Institute of Allergy and Infectious Diseases (NIAID) (www.beiresources.org) at the National Institutes of Health (NIH). Adult *D. andersoni* (Reynold's Creek colony) were used in all *A. marginale* experiments. *I. scapularis* ticks were maintained in an incubator at 23 °C with 85% relative humidity and a 14/10-h light/dark photo-period regimen, while *D. andersoni* ticks were maintained in an incubator at 25 °C with 98% relative humidity and a 12/12-hour light/dark photo-period regimen. Mouse breeding and experiments were performed in strict compliance with guidelines set forth by the NIH (Office of Laboratory Animal Welfare (OLAW) assurance numbers A3200-01, A323-01, A3270-1). Procedures were approved by the Institutional Biosafety (IBC:00002247, HP07-08, DES14-27) and Animal Care and Use (IACUC:0413017, 2014-07941, R15-34) committees at the University of Maryland, Baltimore School of Medicine, University of Maryland, College Park and Yale University of School of Medicine. C3H/HeJ mice (catalogue number 000659) were purchased from Jackson Laboratories. Mice were gender matched and at least 6–10 weeks of age. A low passage infectious isolate of *B. burgdorferi* B31, clone MSK5 (ref. 42) was used. *A. marginale* procedures were approved by the University of Idaho Institutional Animal Care and Use and Biosafety Committees (IACUC, 2013-66; Biosafety, B-010-13). A splenectomized Holstein calf (C82198) was inoculated with *A. marginale*-infected blood and allowed to develop acute infection.

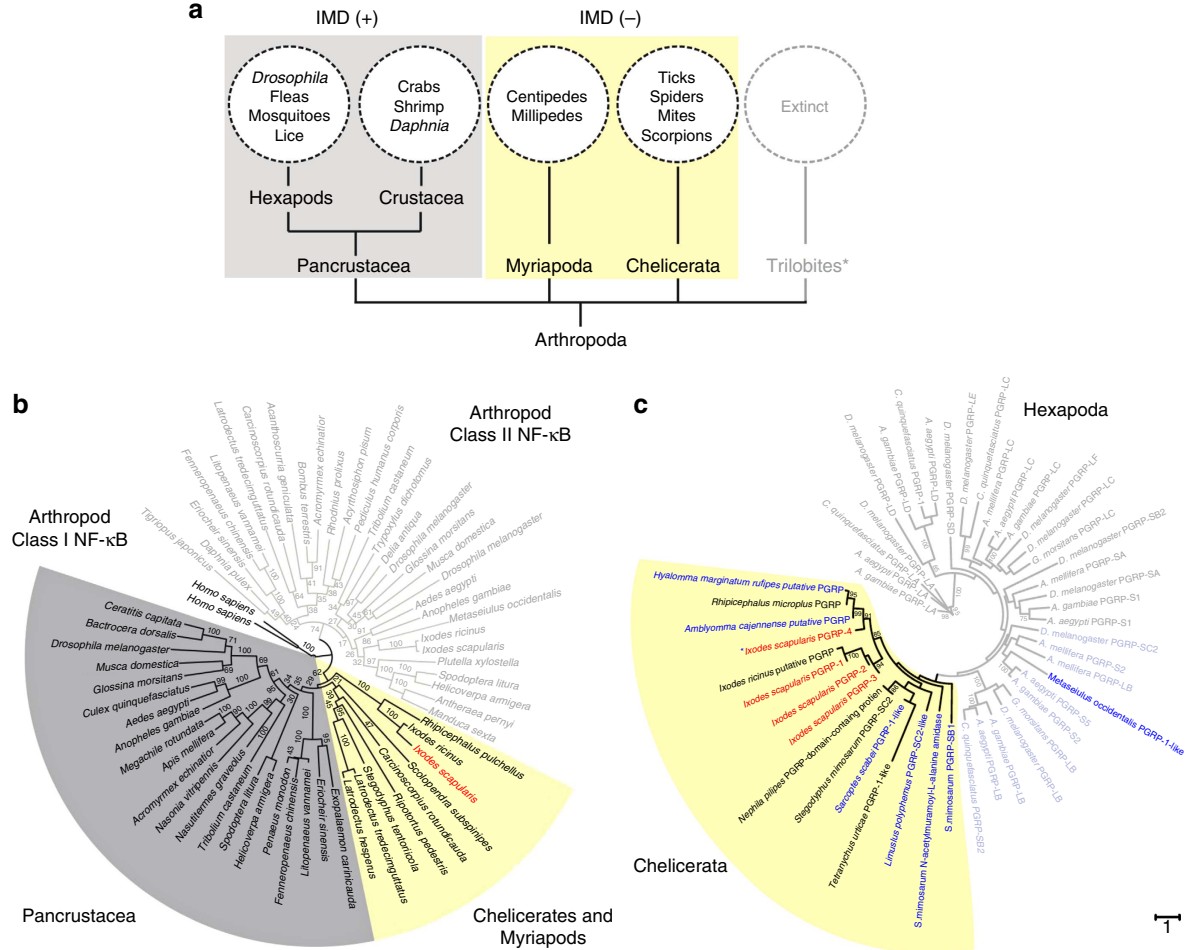

**Figure 7 | The atypical IMD pathway in Chelicerates/Myriapods. (a)** BLAST was used to survey arthropod sequences. The *Drosophila* IMD was used as a query sequence. Confounding factors, such as sparsely populated data matrices, sequence misalignment and biased statistical confidence were removed. *Trilobites are an extinct subphylum. **(b)** The Rel homology domain sequence from *D. melanogaster* Relish was used to search arthropod transcripts for *relish* (class I) and other Rel domain-containing targets (*dorsal* and *dif*; class II) with tBLASTn. Two human NF-κB molecules served as outgroups. **(c)** *I. scapularis* PGRP sequences include PGRP-1: XM_002411731.1, PGRP-2: XM_002433644.1, PGRP-3: XM_002410377.1 and PGRP-4: XM_002413046.1. PGRP-1 was used to search chelicerate proteomes for PGRPs. Bootstrap values greater than or equal to 70 are shown. Yellow shading indicates chelicerate PGRP sequences, with *I. scapularis* PGRPs highlighted in red. Insect PGRPs are coloured gray and light blue. Blue labels and asterisks denote probable amidase activity based on the residues HHC in the active site. **(b,c)** MUSCLE was used to generate the multiple sequence alignment. The maximum likelihood phylogenetic tree was calculated with RAxML and re-sampled 100 times to assess clade support. See also Supplementary Figs 6 and 7.

Culturing for the *A. phagocytophilum* strain HZ and calculations were described elsewhere[43]. Briefly, *A. phagocytophilum* strain HZ was grown in HL-60 cells (ATCC, CCL-240), a human promyelocytic cell line, with Roswell Park Memorial Institute (RPMI) medium supplemented with 10% heat-inactivated FBS and Glutamax (Gibco, 35050-061). Cells were maintained at a concentration between $1 \times 10^5$ and $1 \times 10^6$ ml$^{-1}$ at 37 °C, 5% $CO_2$. Before isolating host-free *A. phagocytophilum*, bacteria were enumerated using a previously reported formula[43]. The percentage of infected cells is multiplied by the average number of microcolonies per cell, termed 'morulae' (5), the average bacteria per morulae (19) and the average amount of bacteria typically recovered from the isolation procedure (50%).

**E. coli, ISE6 and Drosophila melanogaster cell cultures.** *E. coli* cultures[21] were grown overnight in lysogeny broth (LB) supplemented with appropriate antibiotics (ampicillin 100 μg μl$^{-1}$). The tick cell line, ISE6, was a gift from Ulrike Munderloh at the University of Minnesota and was used for all reported *in vitro* tick experiments. Cells were cultured in L15C-300 medium supplemented with 10% heat inactivated fetal bovine serum (FBS, Sigma), 10% tryptose phosphate broth (Difco, 260300), 0.1% bovine cholesterol lipoprotein concentrate (MP Biomedicals, 191476; referred here as L15C-300 complete)[21]. Cells were grown to confluence, as assessed by an inverted light microscope, before either being seeded in 24-well plates (Celltreat, 229124) or split (1:5–1:20) in T25 flasks (Cellstar, 690-160) for culture expansion. To infect ISE6 cells, *A. phagocytophilum* was lysed from HL-60s using a 27-gauge needle, followed by washing with PBS. Infections were allowed to progress for 18 h before cells were collected, unless otherwise stated.

The *Drosophila melanogaster* cell line, S2* was a gift from Neal Silverman at the University of Massachusetts Medical School. Cells were cultured in Schneider's *Drosophila* medium supplemented with 10% heat inactivated fetal bovine serum (FBS, Sigma). For bacterial infection experiments, S2* cells were seeded at $1 \times 10^6$ per well in 24-well plates with 1 μM 20-hydroxyecdysone (Sigma) for 24 h to prime the IMD pathway, as previously reported[44]. Positive controls for IMD and Toll pathway activation were as follows: *E. coli*-derived peptidoglycan (InvivoGen, tlrl-pgnek, 10 μg mL$^{-1}$) stimulation for 6 h and *Staphylococcus aureus* infection for 20 h. For *S. aureus* infections, strain USA300 JE2 (MRSA) was grown overnight at 37 °C on trypticase blood agar plates (5% sheep blood, BD). Single colonies were inoculated into Tryptic Soy Broth (TSB, BD) for overnight liquid culture at 37 °C with 180 rpm shaking. Subcultures were inoculated at 1:100 in TSB in the same conditions for 2.5 h. Optical density (OD600) of the subculture was adjusted to 0.169 in sterile PBS (Gibco). Bacteria were pelleted and the OD-adjusted subcultures were washed in PBS and pelleted. Final pellets were re-suspended in Schneider's medium as described above. Cell cultures were inoculated with *S. aureus* at an MOI of 1,000 or with sterile medium (mock) and cultured for 20 h at 23 °C. For antimicrobial peptide transcript production, infections or incubations were collected after 6 or 20 h corresponding to the reported peak of IMD or Toll pathway activation[33].

**Plasmid construction.** Both *bendless* or *xiap* were amplified by PCR using the primers indicated in Supplementary Table 2. *bendless* was cloned with HindIII and KpnI sites into pCMV/hygro-Negative Control Vector (FLAG-tagged) (Sino Biological, Inc). Similarly, *xiap* was cloned with EcoRI and NotI sites into

pCMV-HA (New MCS) (Received as a gift from Christopher A. Walsh; Addgene plasmid # 32530). Both constructs were confirmed by sequencing. Recombinant Bendless was generated by amplifying the gene from *I. scapularis* cDNA using the indicated primers in Supplementary Table 2 and was cloned into pGEX-6P-2 using *Bam*HI and *Xho*I sites. Recombinant XIAP was produced using the previously reported XIAP expression plasmid[21].

**Mobility shift electrophoresis and western blotting.** Western blotting was performed as previously described[26]. Briefly, proteins were separated by SDS–PAGE and transferred to PVDF membranes. Membranes were blocked with 5% milk in PBS-T (1 × phosphate-buffered saline containing 0.05% Tween 20) for 1–2 h before being incubated with primary antibodies in either 0.25% milk PBS-T or 5% BSA (Bovine Serum Albumin) in PBS (for ubiquitin antibodies) overnight at 4 °C. Primary antibodies are as follows: α-UbK63 (Millipore, 05-1308, 1:1,000), α-UbK48 (Millipore, 05-1307, 1:1,000), α-PanUb (Millipore, MABS486, 1:1,000), α-XIAP (Thermo Scientific, custom, 1:200), α-HA (Sino Biological, 100028-MM10, 1:1,000), α-FLAG (Sigma, F3165, 1:1,000), α-*I. scapularis* Relish (Thermo Scientific, custom, 1:750), α-β-actin (Sigma, A2103, 1:1,000), α-rabbit (Life Technologies, A16023, 1:5,000) and α-mouse (Abcam, AB97046, 1:5,000). All blots were washed and incubated with secondary antibodies for 1 h at room temperature before being visualized with Enhanced chemiluminescence (ECL) western blotting substrate (Thermo Scientific). When necessary, blots were stripped with Western Blot Stripping Buffer (Thermo Scientific). Native gel electrophoresis was performed as previously described[26]. Briefly, 0.2 µg of XIAP was combined with increasing amounts of Bendless, as indicated. Proteins were incubated at room temperature for 4 h followed by native PAGE analysis and immunoblotting.

**ELISA.** 0.2 µg of XIAP was coated into a high-binding 96-well plate with 0.5 M carbonate-bicarbonate (pH 9.5) at 4 °C overnight. Plates were washed with PBS-T and blocked with 10% heat-inactivated FBS in PBS followed by incubation with indicated concentrations of Bendless at room temperature for 1 hour. Equal concentration of purified GST was used as a control. Plates were washed 5X with PBS-T and incubated with the α-UbcH13 (Novus Biologicals, NB100-56357, 1:400) and the α-rabbit IgG-HRP (Abcam; 1:10,000). To evaluate *in vivo* binding, 0.4 µg of lysates from unfed *I. scapularis* nymphs microinjected with either scrambled RNA or siRNA targeting *bendless* were coated onto a 96-well plate with 0.5 M carbonate-bicarbonate (pH 9.5). Increasing concentrations of GST-XIAP were added at room temperature for 1 hour. Equal concentration of purified GST was used as a binding control. Plates were washed 5X with PBS-T and probed with α-GST (Calbiochem, OB03, 1:400) and α-mouse IgG-HRP (Abcam, 1: 10,000). For antibody blocking, 9.1 µM of either Bendless or BSA control were incubated with a mouse monoclonal antibody, α-UbcH13 (Novus Biologicals, H00007334-M01) with indicated titrations at room temperature for 1 h before being added to a 96-well plate coated with 0.2 µg of XIAP. Binding levels were assessed with a polyclonal rabbit antibody, α-UbcH13 (Novus Biologicals, NB100-56357, 1:400). ELISAs were developed with 3,3′,5,5′-tetramethylbenzidine (TMB) (BD Biosciences). Reactions were stopped with 1 M $H_2SO_4$ and the absorbance was measured at 450 nm with a 595-nm correction with the Bio-Rad iMark reader.

**Transfection of HEK293 T cells.** $1 \times 10^6$ HEK293 T cells were seeded into 6-well plates for 18 h followed by 10 µl of Lipofectamine 2,000 (Invitrogen) with 4 µg of pCMV-XIAP-HA and/or pCMV-Bendless-FLAG plasmids in Opti-MEM I Reduced Serum Medium (Invitrogen). The DNA-Lipofectamine 2,000 complex was removed after 5 h and replaced with DMEM, 10% FBS and incubated for 2 days. The transfected cells were lysed in 25 mM Tris-HCl pH 7.4, 150 mM NaCl, 1% NP-40, 1 mM EDTA and 5% glycerol with a protease inhibitor cocktail for 15 min on ice. Whole lysates were centrifuged for 30 min at 4 °C at 12,000 r.p.m. and the supernatants were collected for downstream assays. All HEK293 T cell cultures were validated to be *Mycoplasma* free via PCR.

**Co-immunoprecipitation assay.** The expression of both XIAP-HA and Bendless-FLAG in HEK293 T cells was validated with α-HA (Sino Biological, 100028-MM10, 1: 1,000) and α-FLAG (Sigma, F3165, 1: 1,000). 2 mg of cell lysates were incubated with 80 µL of either cross-linked α-FLAG M2 agarose beads (Sigma, A2220) or α-HA agarose beads (Pierce, 26181) at 4 °C overnight. The beads were washed three times with 50 mM Tris, 150 mM NaCl, pH 7.5. The agarose beads were boiled in 50 µl of 2 × Laemmli buffer for 5 min and analysed via Western blot.

**Recombinant protein and ubiquitylation assays.** *E. coli* cultures transformed with either pGEX-6 P-2-Bendless or pGEX-6 P-2-XIAP[21] were grown to an OD600 of 0.6–0.8 and induced with 0.1 mM of IPTG overnight at 20 °C. Cells were collected at 4,000 × G for 20 min at 4 °C and resuspended in 20 mM Tris pH 8.9, 300 mM NaCl, 5% glycerol. Recombinant Bendless cell pellets were lysed using a low-volume homogenizer (Microfluidics LV1). Soluble lysates were incubated with glutathione agarose affinity purification beads (Thermo Scientific 16100) for 1 hour at room temperature. Recombinant proteins were either eluted with 10 mM of reduced glutathione in 50 mM Tris, 150 NaCl, pH 8 or had the GST-tag cleaved

with 100 U of PreScission Protease in 50 mM Tris-HCL pH 7, 150 mM NaCl, 1 mM EDTA, 1 mM DTT at 4 °C overnight. Recombinant XIAP cell pellets were re-suspended in buffers with pHs ranging from 4 to 10. Samples were sonicated and fractions were separated by centrifugation at 20,000g for 30 min at 4 °C. Affinity purification proceeded as outlined above. An additional buffer exchange step was included using Amicon Ultra Tubes (Millipore, 903024) as well as a size exclusion step to purify GST-tagged XIAP using fast purification liquid chromatography (FPLC). Ubiquitylation assays were performed with the following conditions: reaction buffer (500 mM Tris pH 7.4, 10 mM DTT), Energy R Solution (Boston Biochemical, B-10), 1.2 µg XIAP, 275 ng Ube1 (Boston Biochemical, E-305), 100 ng Bendless, 100 ng Uev1a (Boston Biochemical, E-662), 5 µg wild type ubiquitin or ubiquitin mutants (Boston Biochemical, U-100H, UM-K48R or UMK63R) and resuspended with water in a final volume of 15 µl. Reactions were allowed to proceed for 1 h at 37 °C before being stopped with stop buffer (Boston Biochemical, SK-10).

**Circular dichroism.** To ensure that recombinant XIAP folded properly, the secondary structure of the protein was analysed by circular dichroism (Jasco, Inc.). Protein concentrations were quantified by Bicinchoninic acid assay (BCA) (Pierce, 23225) and diluted to 5 mM for analysis. Data were collected over the spectral range from 190 nm to 260 nm at 1 nm intervals and averaged over three acquisitions. The far-ultra violet circular dichroism spectra showed a prominent minimum at 208 nm, which is consistent with a protein carrying mostly α-helical structures.

**Pull-down assays.** Protein pull-downs assays were carried out with recombinant GST-tagged XIAP crosslinked to glutathione agarose beads with bis(sulfosuccinimidyl)suberate (BS3, ThermoFisher, 21580), following the manufacturer's instructions. $2.5 \times 10^5$ ISE6 cells were sonicated in 20 mM Tris pH 8.9, 150 mM NaCl, 0.01 Triton X-100 with protease inhibitors. Lysates were incubated with cross-linked XIAP for 1 hour at 4 °C. Columns were washed four times and eluted in 20 mM Tris pH 8.9, 150 mM NaCl, 10 mM DTT, 5 mM EDTA, 0.01% Triton X-100 with 'PreScission' protease. Eluted proteins were precipitated using trichloroacetic acid and neutralized with ice cold acetone. 100 µg of protein was digested with trypsin overnight. Samples were quenched with trifluoroacetic acid, desalted and analysed by the University of Maryland, School of Pharmacy Mass Spectrometry Facility.

**Structural modelling.** The tick XIAP sequence was compared with experimentally determined structures from the protein data bank (PDB). Depiction of the least squares structural alignment of the tick (tXIAP) BIR domain and neighbors identified from the Dali server were: 1) cIAP1 BIR3 (PDB:3D9T) - z-score: 18.4; RMSD: 1.2; aligned residues: 95; identity: 35%; 2) hXIAP BIR3 (PDB:3CLX) - z-score: 16.9; RMSD: 1.5; aligned residues: 102; identity: 31%; 3) ML-IAP (PDB: 1OXN) - z-score: 17.6; RMSD: 1.3; aligned residues: 95; identity: 39%; 4) dIAP1 BIR2 (PDB: 1JD6) - z-score: 22.2; RMSD: 0.4; aligned residues: 106; identity: 36%. Structural docking was used to predict protein-protein interactions between XIAP and Bendless. BLAST searches using Bendless (B7PKK7) and XIAP (B7PF95) sequences were evaluated against the PDB. Predicted structures with the highest homology where modelled using the Multiple Mapping Method and Phyre2 programs. The software ZDOCK was used to model the XIAP-Bendless complex. Top ten predictions, which localize to a single interface were shown. Visualization was made by PyMol.

**Protein interactomes.** We acquired the top related proteins interacting with the human XIAP and ML-IAP based on previously observed protein and genetic interactions, pathways and co-localization assays. *I. scapularis* homologues were then identified based on the searches with position-specific iterated (PSI)-basic local alignment search tool (BLAST) and GeneCard. Interactomes were compiled according to GeneMANIA and visualized by Cytoscape. Candidates were grouped according to the functional gene ontology (GO) categories available at the Database for Annotation, Visualization and Integrated Discovery (DAVID).

**iTRAQ.** Data sets from a previous iTRAQ (Isobaric tags for relative and absolute quantitation) experiment deposited on the Dryad repository database (http://dx.doi.org/10.5061/dryad.50kt0) were analysed for proteins of interest.

**RNAi silencing and quantitative reverse transcriptase–PCR.** siRNA and their scrambled controls were synthesized using the primers listed in Supplementary Table 2 and the Silencer siRNA construction Kit (Ambion, AM1620). 3 µg of siRNA or the equivalent scrambled control was transfected into $1 \times 10^5$ ISE6 cells overnight using 5 µl ml$^{-1}$ of lipofectamine 2,000 (Life Sciences, 11668-019). The following day, cells were infected with *A. phagocytophilum*. After 18 h, cells were collected in Trizol (Ambion, 15596018) and stored at −80 °C. RNA was extracted using the PureLink RNA Mini Kit (Ambion, 12183025). cDNA was synthesized from 500 ng of RNA with the Verso cDNA Synthesis Kit (ThermoFisher, AB-1453). Gene silencing and bacterial burden were assessed by quantitative

reverse transcriptase–PCR using the primers described in Supplementary Table 2. All data were expressed as means ± s.e.m.

**I. scapularis microinjections.** Tick microinjections were done, as previously described[21], with approximately 10 ng of siRNA or equivalent scrambled controls. 10 μl microdispensers (Drummond Scientific) were drawn to fine point needles using a micropipette puller (Sutter Instruments). Microinjections were performed using a micromanipulator (Narishige, Tokyo) connected to a Nanojet microinjector (Drummond Scientific). For each group, 20 ticks were microinjected with either siRNA or scRNA and were then allowed to rest for 3–24 h before being placed onto infected mice. Each group of 20 was placed on a single infected mouse. Ticks were allowed to feed to repletion and were then collected for analysis.

**Lipid identification and priming assays.** Host-free *A. phagocytophilum* was isolated and lipids were extracted using methods that were previously described[45]. Briefly, triplicate cell pellets were prepared from infected and uninfected cultures. Cells were re-suspended in the water volume of the Bligh and Dyer single-phase extraction solution followed by the addition of methanol and chloroform for a total lipid extraction. Total lipids were dried under a gentle stream of nitrogen and reconstituted in a 2:1 (v:v) mixture of chloroform:methanol at equal volumes. For analysis by MALDI-TOF, 1 μL was spotted followed by 1 μL of norharmane matrix at 20 mg ml$^{-1}$ in the same diluent[46]. These triplicate samples, along with triplicate uninfected controls, were analysed in at least technical duplicate by MALDI-TOF mass spectrometry in negative mode (Bruker Daltonics Autoflex Speed MALDI-TOF; Billerica, MA) and identified according to the lipid metabolites and pathways strategy nomenclature (LIPID MAPS)[47]. To identify relative changes between the uninfected control and *A. phagocytophilum*, all mass channels from $m/z$ 700–900 exceeding a signal:noise ratio > 6 were exported and analysed further. A cluster of PG ions were upregulated at least 2-fold (by S:N ratio comparison) in *A. phagocytophilum* samples, dominated by a cluster of PG species containing 34 acyl carbons in unsaturated, mono-, and di-unsaturated configurations. Specifically, $m/z$ 747.5 was identified (putative identity assigned as PG 34:1, commonly observed as palmitoyl (16:0)-oleoyl(18:1)-PG, POPG)[48,49] as an ion of interest due to the dramatic increase in relative abundance compared with uninfected cells. Additionally, PG 34:0 and PG 34:2 were unique to *A. phagocytophilum* (Supplementary Table 3). Supplementary Table 3 highlights ions exclusive in both conditions and includes the 3 ions exceeding 2-fold detection over uninfected cells predicted to be even-carbon chain PGs. Exact masses are given from LIPID MAPS for error calculation. All organic solvents and MALDI reagents were sourced from Sigma-Aldrich (St Louis, MO).

For priming experiments, 0.01–1 ng of reference lipids (Avanti Polar Lipids, 840457, 800815 and 850445) were diluted into the tick cell culture and were added to previously seeded cells. Stimulation proceeded for 6 h, corresponding to the height of IMD pathway activity[33]. Media was then replaced with media containing *A. phagocytophilum*. Infection progressed for 18 h before samples were collected. For sequential silencing and priming experiments, targeted RNAi silencing was performed, as described earlier, before medium containing lipofectamine/RNAi was removed and replaced with lipid-containing medium.

To examine whether lipids affected *A. marginale* (St Maries strain) infection, we injected groups of unfed, adult male *D. andersoni* with individual lipids or a control. On the calf reaching a bacteremia of 1.6% (16 days post-infection; packed cell volume = 36%), five cohorts of 150 adult male *D. andersoni* were injected, as previously described[50] with either 1 ng of POPG, 1 ng of PODAG, 1 ng of MPPC, or 1 μl of chloroform/MeOH (lipid diluent control) diluted in Hanks buffered saline solution per tick and were immediately placed on the calf. Ticks were allowed to feed for six days and were then removed and held at 26 °C for seven days. Midguts from individual ticks were assessed for *A. marginale* infection levels using quantitative PCR and the primers described in Supplementary Table 2. The calf was killed the same day ticks were removed and had a final bacteremia = 30.4% and packed cell volume = 21%. 95–100% of injected ticks were recovered from their respective patches.

**Relish antibody production and immunoblot.** A polyclonal antibody was raised against the *I. scapularis* protein Relish. The protein sequence was empirically determined by amplifying *relish* from ISE6 cDNA using the primer combination Isc_Relish 5′ and 3′ (Supplementary Table 2). This resulting amplicon was sequenced and used to predict an amino acid sequence. Based on this, the peptide sequence REDGRATFPSMSIVFQQKK, drawn from the Rel-homology domain (RHD) portion of *I. scapularis* Relish, was synthesized and used to raise specific rabbit polyclonal antibodies (Pierce Antibodies, custom services). For immunoblots, ISE6 cell cultures were grown and lysed with radioimmunoprecipitation assay buffer (RIPA, Teknova, R3792) supplemented with protease inhibitors (Pierce, 88665).

**Phylogenetic analysis of *imd*, *relish* and PGRPs.** *Imd* was searched for in Arthropoda using the Basic Local Alignment Search Tool (BLAST) available through NCBI. All available arthropod genomes (tBLASTn), transcriptomes (tBLASTn) and proteomes (PSI-BLAST) were mined using the *D. melanogaster* IMD amino acid sequence. This BLAST analysis does not reflect overrepresentation

of any genus within the subphylum. Data sets available for each subphylum are as follows: Hexapoda: 100 genomes, 231 transcriptomes. Crustacea: 3 genomes, 41 transcriptomes. Myriapoda: 1 genome, 18 transcriptomes. Chelicerata: 13 genomes, 44 transcriptomes.

The Rel homology domain sequence from *D. melanogaster* Relish was used to search arthropod transcripts for *relish* (class I) and other Rel homology domain-containing targets (*dorsal* and *dif*; class II) with tBLASTn. A multiple sequence alignment method with reduced time and space complexity (MUSCLE)[51] was used to generate the multiple sequence alignment. The maximum likelihood phylogenetic tree was calculated with RAxML[52] and resampled 100 times to assess clade support. The phylogenetic tree was visualized and annotated with the Interactive Tree of Life tool[53]. Two human NF-κB transcripts served as outgroups.

Annotated PGRP protein sequences for *Aedes aegypti*, *Apis mellifera*, *Anopheles gambiae*, *Culex quinquefasciatus*, and *D. melanogaster* were downloaded from NCBI. *I. scapularis* PGRP-1 was used to search chelicerate proteomes for PGRPs. MUSCLE was used to align the protein sequences. The maximum likelihood phylogenetic tree was calculated with RAxML and resampled 100 times to assess clade support. Bootstrap values greater than or equal to 70 are shown.

**Statistical analysis.** Sample sizes were chosen based on methods that have previously been reported in the literature and what has historically been appropriate to achieve statistical power[21,26,54–58]. *In vitro* experiments were performed with 3–5 replicates. *In vivo* experiment involved the use of 10-20 ticks. Data were expressed as means ± s.e.m. and analysed with either the unpaired Student's *t*-test or one-way analysis of variance. Calculations and graphs were made by using GraphPad Prism version 6.0. $P < 0.05$ was considered statistically significant.

**Data availability.** Protein structural data that support the findings of this study have been deposited in the Protein Data Bank with the primary accession codes 3D9T (cIAP1 BIR3), 3CLX (hXIAP BIR3), 1OXN (ML-IAP), and 1JD6 (1JD6). iTRAQ data referenced in this study are available in the Dryad Digital Repository with the identifier http://dx.doi.org/10.5061/dryad.50kt0[25]. Sequence data for Bendless and XIAP interactions referenced in this study are available in Uniprot with the accession codes B7PKK7 and B7PF95. Sequence date referenced in Supplementary Tables 1 and 2 are available in UnitProt or the National Center for Biotechnology Information with accession codes provided in those Supplementary Tables. Other data that support the findings of this study are available from the corresponding author on request.

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

## Acknowledgements

We acknowledge Kimberly Stephens, Gregor Blaha (University of California, Riverside) and Sukanya Narasimhan (Yale University) for technical assistance; Ulrike Munderloh (University of Minnesota) for providing tick ISE6 cells; Jon Skare (Texas A&M Health Science Center) for providing the *B. burgdorferi* B31 strain, clone MSK5; Neal Silverman (University of Massachusetts Medical School) for providing S2* *Drosophila* cells; the Core facilities at the University of Maryland, Baltimore for services related to circular dichroism and proteomics. This work was supported by the National Institutes of Health (R01 AI093653 and R01AI116523 to J.H.F.P.) and the University of Maryland, Baltimore School of Medicine. E.E.M. was a trainee under the Institutional Training Grant T32AI007540 from the National Institute of Allergy and Infectious Diseases. The content is solely the responsibility of the authors and does not necessarily represent the official views of the National Institute of Allergy and Infectious Diseases or the National Institutes of Health.

## Author contributions

J.H.F.P. and D.K.S. conceived and designed the study; D.K.S. and L.J.B. performed most experiments, assisted by X.W. (ELISAs, mobility shift electrophoresis, HEK293 T cell pull downs); A.S.O.C. purified recombinant XIAP and Bendless, and performed the polyubiquitylation assays; E.E.M. performed phylogenetic analyses and constructed figures of arthropod NF-κB and PGRP sequences; V.M.B. designed and constructed the HEK293 T cell expression vectors; H.L.H. cloned the pGEX-6P-2 Bendless plasmid; G.A.S. performed structural modelling and protein purification; L.L., K.D., K.E.R. and A.A.S. performed the tick experiments; A.J.S. and R.K.E. performed the lipid analysis; G.A.S., E.F., U.P., M.V., J.d.l.F., M.W.U. and E.J.S. contributed reagents and analytic tools; J.H.F.P., L.J.B. and D.K.S. analysed the data; J.H.F.P. and D.K.S. wrote the manuscript. All authors discussed the results and commented on the manuscript.

## Additional information

**Competing financial interests**: The authors declare no competing financial interests.

**How to cite this article**: Shaw, D. K. *et al.* Infection-derived lipids elicit an immune deficiency circuit in arthropods. *Nat. Commun.* **8**, 14401 doi: 10.1038/ncomms14401 (2017).

**Publisher's note**: 

