## [Peer Review File · Nature Communications]

Reviewers' comments:

Reviewer #1

Expert in IMD

(Remarks to the Author):

The manuscript NCOMMS-16-11105 titled "Infection-Derived Lipids Elicit a Novel Immune Deficiency Circuitry in Arthropods" by Shaw et al. describes the tick equivalent of the Imd pathway, which appears somewhat different to the pathway in insects and crustaceans. Although the tick and insect Imd pathways share many of the downstream components, ticks lack the DAP-type peptidoglycan receptors PGRP-LC and -LE, the adaptor protein Imd, and the death domain protein FADD. Instead, the tick immune response is mediated by the E3 ligase XIAP. In this manuscript, the authors argue that XIAP interacts with the E2 ligase Bendless, and that infection-derived lipids can trigger both the tick and insect Imd pathways. Injecting ticks with these lipids prior to microbial exposure resulted in lower bacterial burden in both *I. scapularis* and *D. andersoni* ticks.

The findings in this paper, especially the immune-stimulatory function of infection-derived lipids, are very interesting. However, there are several points in the proposed model that the authors need to clarify. Please see my detailed comments below.

In Figure 1 the authors describe a model of the structure of tXIAP. Although a cartoon of tXIAP can be found in a previous publication, it would not harm to remind the reader that, unlike mammalian XIAPs, the tick XIAP only has one BIR domain. Also, what do the yellow and cyan residues represent in Fig 1a?

In Figure 3 the authors do not show the actual data for Bendless being pulled down by recombinant XIAP. To show direct interaction of these two proteins, the authors need to provide the MS data that they are referring to, or perform another pull-down with tagged proteins. In Figure 3d I find it interesting that in the tick cell line ISE6 Relish is cleaved upon stimulation with DAP-PGN. Wouldn't that suggest that there is a receptor for DAP-PGN in tick cells after all? If not, how is this response occurring? How pure are the PGN ligands being used? In Figure 3f I am not sure what the siRNA treatment is for Bendless-Uev1a. Two siRNAs knocking down both Uev1a and Bendless? If yes, then how is the RNAi efficiency measured for both of them since there is only one column? This also applies for Figures 4 and 6. In Figure 3e-h some of the fold changes have been marked with decrease in % and some as fold change. I find this confusing in Figures 4, 5, and 6 too. Also, the molecular weights should be marked on Figure 3d. The authors state in materials and methods that the error bars represent SEM, but this should be stated in the figure legends throughout the manuscript, as well as the number of replicates in each experiment. Overall the data presentation is very sloppy, greatly detracting for the manuscript.

In Figure S4 the authors show that *B. burgdorferi* and *A. phagocytophilum* trigger the Imd pathway activation despite lacking DAP-type PGN, but elicit no response via the Toll pathway in *Drosophila* S2* cells. If these bacteria have no DAP-PGN, what is the mechanism for Imd pathway activation? This data, while very interesting, is further compromised by a lack of a positive control for Toll pathway activation. Also confusing matters is that the scales on y-axes in S4a and S4b are different. How fast is the IM1 induction in S2* cells? The Toll pathway activation is supposed to be slower than the Imd pathway response, and I think the kinetics in this assay should be controlled by showing that something activates the Toll pathway in 6h. Also, in Figure S4b values on y-axis are confusing. What does dipterin (or IM1) / 10^3 rp49 mean? Why is the baseline for non-stimulated cells 2000? In S4b, *Anaplasma* seems to induce dipterin expression less than 2x. How does this compare to S4a, where the induction is more than 300x?

Figure 5 shows that POPG and PODAG (34:1) lipids can elicit the Imd pathway response in both fly and tick cells. Are all the lipid-stimulated samples in Figure 5b compared to the non-stimulated

sample? If yes, how come the middle red column (MPPC, 0.1ng?) is not significantly different? How pure are these lipids? Free of PGN? As mentioned above, error bars and N need to be indicated in the figure legend. I understand that antimicrobial peptides are not known in ticks and therefore their induction cannot be used as a read-out, but could the authors need to more directly show that the lipids really trigger the tick Imd pathway? Is Relish for example cleaved in response to POPG and PODAG, but not MPPC?

In Figure S5 the authors show that tick PGRPs don't play a role in the Imd pathway activation by measuring the effect of silencing each PGRP separately or together and measuring bacterial loads in ISE6 cells. To me it looks like all except PGRP2 have been normalized to 1. Is there a reason for this? Also, RNAi efficiency in S5e is not shown. This is a critical control, and without it the authors cannot conclude anything.

Reviewer #2

Expert in pathogenesis of spirochetal infections
(Remarks to the Author):

The article by Shaw et al. "Infection-Derived Lipids Elicit a Novel Immune Deficiency Circuitry in Arthropods" describes an atypical IMD pathway in ticks that is activated by lipids (POPG and PODAG) but not by peptidoglycan. This alternative IMD pathway lacks peptidoglycan receptors, IMD and FADD molecules but in turn has an ubiquitin ligase (XIAP) that plays a central role by interacting with the E2 conjugating enzyme, Bendless. In this manuscript, the authors prove that the IMD pathway in ticks is quite different from other arthropods and it is activated by different molecules. This has evolutionary implications that may help to better understand the interaction of human and animal pathogens and their tick vectors.

Below there are some comments that should be considered and addressed.

1. The association between XIAP and Bendless is not very convincing since the only empirical data provided for this interaction is a pulldown assay. The authors may want to consider to strengthen their claim by including additional experiments to prove that a direct interaction of these two proteins actually exists. In the absence of additional experiments, the interaction claim should be markedly re-assessed and the conclusions toned-down. In addition, the in silico data, mostly protein docking, is only displayed as an image but with no analytical data to assess the likelihood of this interaction. The image alone is not very useful.

2. Lines 143-145 and Figure 3d need clarification. The authors stated in the text "Similarly, A phagocytophilum induced the appearance of Rel-N as early as 1 min (peaking at 5 minutes) post stimulation of the I. scapularis tick cell line ISE6 (figure 3d)". However, in figure 3d Rel-N is more abundant at 1 minute and seems to decrease over time (5 min).

3. Why Bendless and uev1a are silenced simultaneously? This is not obvious to the reader.

4. Only Bendless-uev1a were silenced in vivo for Anaplasma. The same should be done with Relish and Caspar to prove that this is not an in vitro artifact.

5. Figure 5B. The values of dipteracin for the different concentrations of MPPC used in this experiment are higher than the dipteracin value for the lowest concentration of PODAG but only the latter is statistically significant. Please clarify.

6. Figure 5c-d. Pre-incubation with 1ng of PODAG confers protection to ISE6 cells (ticks) but not to S2 cells (Drosophila). Could it be possible that PODAG elicit a higher response in tick cells than in Drosophila cells?

Reviewer #3

Tech. reviewer for lipidomics

(Remarks to the Author):

I have examined the manuscript from the point of view of a lipid expert only. The authors briefly state that the lipid composition of host free *A.phagocytophilum* was investigated by MALDI TOF/MS and that phosphatidylglycerol was the main lipid component.

The experiments performed to test whether exogenous POPG, PODAG and MPPC could stimulate humoral immun pathways look ok to me.

However a description of the methods of lipid analysis (lipid extraction method, matrix used in mass spectrometry analysis) is missing and should be added; if possible the lipid profile of the microorganism could be included in the manuscript (only as supplementary information of course) also because, to my knowledge, very little information on lipids of *A.phagocytophilum* is available in the literature.

Reviewer #1 (Expert in IMD):

The manuscript NCOMMS-16-11105 titled "*Infection-Derived Lipids Elicit a Novel Immune Deficiency Circuitry in Arthropods*" by Shaw *et al.* describes the tick equivalent of the Imd pathway,

which appears somewhat different to the pathway in insects and crustaceans. Although the tick and insect Imd pathways share many of the downstream components, ticks lack the DAP-type peptidoglycan receptors PGRP-LC and -LE, the adaptor protein Imd, and the death domain protein FADD. Instead, the tick immune response is mediated by the E3 ligase XIAP. In this manuscript, the authors argue that XIAP interacts with the E2 ligase Bendless, and that infection-derived lipids can trigger both the tick and insect Imd pathways. Injecting ticks with these lipids prior to microbial exposure resulted in lower bacterial burden in both *I. scapularis* and *D. andersoni* ticks. The findings in this paper, especially the immune-stimulatory function of infection-derived lipids, are very interesting. However, there are several points in the proposed model that the authors need to clarify. Please see my detailed comments below.

1. In Figure 1 the authors describe a model of the structure of tXIAP. Although a cartoon of tXIAP can be found in a previous publication, it would not harm to remind the reader that, unlike mammalian XIAPs, the tick XIAP only has one BIR domain. Also, what do the yellow and cyan residues represent in Fig 1a?

We thank Reviewer# 1 for this helpful suggestion. The former Figure 1 in the original submission has now been moved to Supplementary Figure 3. We have added an additional schematic to highlight the domain differences between human, mouse, *Drosophila* and tick homologues, which can be found in Supplementary Figure 3b. In Supplementary Figure 3a, the BIR and RING structures of XIAP highlights residues in stick representation. Cyan and yellow side chains represent conserved amino acids that are involved in Zn atom coordination. The superimposition of cIAP1 (green) shows the absence of a known CARD or UBA domain in the *I. scapularis* XIAP protein (red).

2. In Figure 3 the authors do not show the actual data for Bendless being pulled down by recombinant XIAP. To show direct interaction of these two proteins, the authors need to provide the MS data that they are referring to, or perform another pull-down with tagged proteins.

In the revised manuscript, we now show through 6 distinct approaches that XIAP and Bendless directly and specifically interact:

- 1) We demonstrate that Bendless and XIAP specifically interact through the use of mobility shift assays. Recombinant XIAP was incubated with increasing concentrations of purified recombinant Bendless. We display that with increasing amounts of Bendless, a band representing XIAP is shifted to a higher molecular weight when analyzed by electrophoresis under non-denaturing conditions (Figure 2c).**
- 2) We show through the use of ELISA with XIAP as a bait protein that increasing concentrations of Bendless results in binding (Figure 2d).**
- 3) The interaction between XIAP and Bendless can be specifically blocked in an ELISA-based assay when Bendless is pre-incubated with a monoclonal antibody against the human Bendless homologue, Ubch13 (Figure 2e).**
- 4) We show that recombinant XIAP and Bendless, when used in combination, are able to carry out enzymatic reactions and produce K63-, but not K48-, polyubiquitylation. This proves that XIAP and Bendless are not simply interacting physically, but they are specifically pairing together to catalyze polyubiquitylation (Figure 2f).**
- 5) We employed a HEK293 T cell transfection system to demonstrate XIAP-Bendless interactions inside the cell. In this scenario, both XIAP and Bendless were cloned into expression vectors and transfected into HEK293 T cells. Immunoblotting against both the FLAG tag (Bendless-FLAG) and the HA tag (XIAP-HA) demonstrated robust protein expression. Importantly, immunoprecipitation using these affinity tags as targets revealed that Bendless specifically pulled down XIAP and vice versa (Figure 2g).**
- 6) Finally, to provide evidence for XIAP-Bendless interactions *in vivo*, we extracted protein from *I. scapularis* nymphs microinjected with either siRNA targeting *bendless* or with a scrambled control. The protein extracts were used as bait in an ELISA-based assay. Dose-dependent binding of purified XIAP to tick protein extracts was significantly higher in**

control ticks treated with scrambled RNA when compared to ticks in which *bendless* was knocked down (Figure 2h).

3. In Figure 3d, I find it interesting that in the tick cell line ISE6, Relish is cleaved upon stimulation with DAP-PGN. Wouldn't that suggest that there is a receptor for DAP-PGN in tick cells after all? If not, how is this response occurring? How pure are the PGN ligands being used?

Former Fig. 3d has been expanded to include stimulation with lipids and is now Fig. 5f-g.

The *E. coli* K12 peptidoglycan ligand we used in our assays was purchased from InvivoGen (catalogue #: ttrl-pgnek) and is certified to be free of endotoxins. We are therefore confident that the response observed in tick cells is specific to peptidoglycan and not from other impurities. There are 4 encoded PGRPs within the tick genome and none are predicted to be transmembrane (Gulia-Nuss *et al.* 2016, *Nat Commun*; Palmer and Jiggins 2015, *Mol Biol Evol*). This deviates from what is described in the *Drosophila* IMD pathway. Previous *Drosophila* studies have reported that intracellular receptors assist in the activation of the IMD pathway and can have co-receptor functioning to assist transmembrane PGRPs. Tick cells may be capable of responding to DAP-PGN, particularly, if invaded by intracellular bacteria carrying peptidoglycans. Presently, however, there is no evidence that this occurs in ticks via a transmembrane PGRP-IMD axis. We have elaborated on this interesting point in the discussion section (Pages 14-15, Lines 312-318)

4. In Figure 3f I am not sure what the siRNA treatment is for Bendless-Uev1a. Two siRNAs knocking down both Uev1a and Bendless? If yes, then how is the RNAi efficiency measured for both of them since there is only one column? This also applies for Figures 4 and 6.

We apologize for not being clear and have now included graphs noting the silencing levels of both *bendless* and *uev1a* in all pertinent figures. The former Figure 3f is now Figure 3b. Figures 4a, 4d and 6c have also been modified to include the separate silencing levels for both *bendless* and *uev1a*.

5. In Figure 3e-h some of the fold changes have been marked with decrease in % and some as fold change. I find this confusing in Figures 4, 5, and 6 too.

All transcript levels and bacterial loads have now been converted to fold change for consistency. Revisions were made to Figures 3, 4, 5d, 5e, and 6. (Former Figure 3e-h are now Figure 3a-d).

6. Also, the molecular weights should be marked on Figure 3d.

This concern has been noted and is now indicated in Figure 5f-g (formerly Figure 3d in the original submission). Our Relish antibody detects a cleaved product in stimulated cells that migrates to approximately 41 kDa, which is in good agreement with the predicted 39 kDa protein encoded by the *I. scapularis relish*: XM_002434459.1 (Gulia-Nuss *et al.* 2016, *Nat Commun*).

7. The authors state in materials and methods that the error bars represent SEM, but this should be stated in the figure legends throughout the manuscript, as well as the number of replicates in each experiment. Overall the data presentation is very sloppy, greatly detracting for the manuscript.

We apologize for not being clear. We have now included SEM and replicate numbers in the figure legends and in multiple places within the methods section of the manuscript.

8. In Figure S4 the authors show that *B. burgdorferi* and *A. phagocytophilum* trigger the Imd pathway activation despite lacking DAP-type PGN, but elicit no response via the Toll pathway in *Drosophila* S2* cells. If these bacteria have no DAP-PGN, what is the mechanism for Imd pathway activation?

In this study, we provide evidence that infection-derived lipids, POPG and PODAG, are stimulating the IMD pathway (Figures 5 and 6). This is true for both *Drosophila* cells (Fig. 5b-d) and tick cells (Fig. 5e-g). The mechanism of activation does not signal through *I. scapularis* peptidoglycan recognition receptors that are predicted in the genome (all of which are either extracellular or intracellular; Supplementary Figure 6) or the adapter molecule, IMD, for which

the pathway is named (Figure 7 and Supplementary Figure 7). Instead, signaling is reliant on the E3 ubiquitin ligase, XIAP (Figures 3a, and 6a-b), the E2 conjugating enzymes, Bendless and Uev1a (Fig. 3b, 4a, 4d and 6c-d), the positive transcriptional regulator, Relish (Fig. 3c, 4b, 4e, and 5f-g), and the negative regulator of the IMD pathway, Caspar (Fig. 3d, 4c and 4f).

9. This data, while very interesting, is further compromised by a lack of a positive control for Toll pathway activation. Also confusing matters is that the scales on y-axes in S4a and S4b are different. How fast is the IM1 induction in S2* cells? The Toll pathway activation is supposed to be slower than the Imd pathway response, and I think the kinetics in this assay should be controlled by showing that something activates the Toll pathway in 6h.

We have now included a positive control to show Toll pathway stimulation with S2* cells in Fig. 5b and have standardized the y-axes on Supplementary Figure 5 for consistency (former Supplementary Figure 4). Lysine-type peptidoglycan is known to activate the Toll pathway (Leulier et al. 2003. *Nature Immunology*). We have now included data with *Drosophila* S2* cells infected with a lysine-type peptidoglycan-containing bacteria, *Staphylococcus aureus*. We show that at 20 hours post-infection, only *S. aureus* upregulates the Toll-specific antimicrobial peptide, *im1*, while *A. phagocytophilum*, POPG, PODAG and MPPC do not (Fig 5b).

10. Also, in Figure S4b values on y-axis are confusing. What does dipterucin (or IM1) /10³ rp49 mean? Why is the baseline for non-stimulated cells 2000? In S4b, *Anaplasma* seems to induce dipterucin expression less than 2x. How does this compare to S4a, where the induction is more than 300x?

We apologize for the unclear y axis annotation and have converted data to fold change for ease of interpretation. The original baseline value for non-stimulated cells was high due to pre-treatment with ecdysone, which is necessary to prime S2* cells prior to stimulating with an IMD pathway agonist *in vitro*. This has been noted by Rus et al. 2013 *EMBO J*. The induction power of each stimulant will vary, depending on the agonist. This is observed with purified DAP-PGN by itself (Fig. 5c) versus other non-DAP-PGN containing bacteria such as *A. phagocytophilum* and *B. burgdorferi* (Fig. 5c, Supplementary Figures 5a and 5c). Some pathogens may be more potent stimulators of the IMD pathway, while others subvert the immune response altogether. Because multiple agonists have been used during this study, it is difficult to perform a direct comparison, as they are molecularly and structurally distinct.

11. Figure 5 shows that POPG and PODAG (34:1) lipids can elicit the Imd pathway response in both fly and tick cells. Are all the lipid-stimulated samples in Figure 5b compared to the non-stimulated sample? If yes, how come the middle red column (MPPC, 0.1ng?) is not significantly different? How pure are these lipids? Free of PGN?

All lipid-stimulation assays are compared directly to non-stimulated samples. In Figure 5c (formerly Figure 5b in the original submission), MPPC samples did not vary significantly from the non-stimulated cells when compared using a one-tailed student's t-test. The P values for each MPPC sample are as follows:

1 ng MPPC vs. non-stimulated: p-value = 0.0576
0.1 ng MPPC vs. non-stimulated: p-value = 0.0729
0.01 ng MPPC vs. non-stimulated: p-value = 0.0760

These lipids were synthesized by Avanti Polar Lipids (catalogue numbers 840457, 800815 and 850445) and have been certified to be free of contaminants as per the product specification sheets.

12. I understand that antimicrobial peptides are not known in ticks and therefore their induction cannot be used as a read-out, but could the authors need to more directly show that the lipids really trigger the tick Imd pathway? Is Relish for example cleaved in response to POPG and PODAG, but not MPPC?

We have included additional experimentation to address this question. We have included Western blots evaluating the cleavage of Relish in tick cells following stimulation with POPG, PODAG and MPPCs at both 1 and 5 minutes (Figure 5f) and with increasing concentrations (Figure 5g). For both of these experiments, POPG and PODAG, but not MPPC, lead to the cleavage of Relish (Figure 5f-g).

13. In Figure S5 the authors show that tick PGRPs don't play a role in the Imd pathway activation by measuring the effect of silencing each PGRP separately or together and measuring bacterial loads in ISE6 cells. To me it looks like all except PGRP2 have been normalized to 1. Is there a reason for this? Also, RNAi efficiency in S5e is not shown. This is a critical control, and without it the authors cannot conclude anything.

scPGRP2 has now correctly been normalized to 1 (Supplementary Fig. 6b). We have now included the silencing levels for the combined knockdown in Supplementary Fig. 6e.

Reviewer 2 (Expert in pathogenesis of spirochetal infections)

The article by Shaw *et al.* "*Infection-Derived Lipids Elicit a Novel Immune Deficiency Circuitry in Arthropods*" describes an atypical IMD pathway in ticks that is activated by lipids (POPG and PODAG) but not by peptidoglycans. This alternative IMD pathway lacks peptidoglycan receptors, IMD and FADD molecules but in turn has a ubiquitin ligase (XIAP) that plays a central role by interacting with the E2 conjugating enzyme, Bendless. In this manuscript, the authors prove that the IMD pathway in ticks is quite different from other arthropods and it is activated by different molecules. This has evolutionary implications that may help to better understand the interaction of human and animal pathogens and their tick vectors. Below there are some comments that should be considered and addressed.

1. The association between XIAP and Bendless is not very convincing since the only empirical data provided for this interaction is a pulldown assay. The authors may want to consider to strengthen their claim by including additional experiments to prove that a direct interaction of these two proteins actually exists. In the absence of additional experiments, the interaction claim should be markedly re-assessed and the conclusions toned-down. In addition, the *in silico* data, mostly protein docking, is only displayed as an image but with no analytical data to assess the likelihood of this interaction. The image alone is not very useful.

In the revised manuscript, we now show through 6 distinct approaches that XIAP and Bendless directly and specifically interact:

- 1) We demonstrate that Bendless and XIAP specifically interact through the use of mobility shift assays. Recombinant XIAP was incubated with increasing concentrations of purified recombinant Bendless. We display that with increasing amounts of Bendless, a band representing XIAP is shifted to a higher molecular weight when analyzed by electrophoresis under non-denaturing conditions (Figure 2c).**
- 2) We show through the use of ELISA with XIAP as a bait protein that increasing concentrations of Bendless results in binding (Figure 2d).**
- 3) The interaction between XIAP and Bendless can be specifically blocked in an ELISA-based assay when Bendless is pre-incubated with a monoclonal antibody against the human Bendless homologue, UbcH13 (Figure 2e).**
- 4) We show that recombinant XIAP and Bendless, when used in combination, are able to carry out enzymatic reactions and produce K63-, but not K48-, polyubiquitylation. This proves that XIAP and Bendless are not simply interacting physically, but they are specifically pairing together to catalyze polyubiquitylation (Figure 2f).**
- 5) We employed a HEK293 T cell transfection system to demonstrate XIAP-Bendless interactions inside the cell. In this scenario, both XIAP and Bendless were cloned into expression vectors and transfected into HEK293 T cells. Immunoblotting against both the FLAG tag (Bendless-FLAG) and the HA tag (XIAP-HA) demonstrated robust protein**

expression. Importantly, immunoprecipitation using these affinity tags as targets revealed that Bendless specifically pulled down XIAP and vice versa (Figure 2g).

- 6) Finally, to provide evidence for XIAP-Bendless interactions *in vivo*, we extracted protein from *I. scapularis* nymphs microinjected with either siRNA targeting *bendless* or with a scrambled control. The protein extracts were used as bait in an ELISA-based assay. Dose-dependent binding of purified XIAP to tick protein extracts was significantly higher in control ticks treated with scrambled RNA when compared to ticks in which *bendless* was knocked down (Figure 2h).

2. Lines 143-145 and Figure 3d need clarification. The authors stated in the text "Similarly, *A. phagocytophilum* induced the appearance of Rel-N as early as 1 min (peaking at 5 minutes) post-stimulation of the *I. scapularis* tick cell line ISE6 (figure 3d)". However, in figure 3d Rel-N is more abundant at 1 minute and seems to decrease over time (5 min).

We thank Reviewer 2 for bringing this point to our attention. We have since modified the text in the manuscript on Page 12, Lines 245-247 to read "Similarly, both *A. phagocytophilum* and infection-derived lipids, POPG and PODAG, induced the appearance of Rel-N as early as one-minute post-stimulation of the *I. scapularis* tick cell line ISE6 (Fig. 5f)" (former Figure 3d has now been expanded to include POPG, PODAG and MPPC and is now Fig. 5f).

3. Why Bendless and *uev1a* are silenced simultaneously? This is not obvious to the reader.

We apologize for not making our justification for dual silencing clear and have incorporated an explanation in the text (Page 9, lines 174-178). Because the interactome and *in vitro* binding analyses show that XIAP and Bendless molecularly interact, we knocked down the heterodimeric E2-conjugating enzyme complex of the IMD pathway (Buchon *et al.* 2014, *Nat Rev Immunol*; Kleino and Silverman, 2014, *Dev Comp Immunol*) consisting of both Bendless and Uev1a. In the mammalian system, Uev1a binds to and activates the human homologue of Bendless, UbcH13, to carry out polyubiquitylation (Hofman and Pickart, 1999, *Cell*; Deng *et al.*, 2000, *Cell*). Previously our group demonstrated that the tick XIAP carries out polyubiquitylation with the human UbcH13-Uev1a complex (Severo *et al.* 2013, *J Infect Dis*). Due to the degree of protein sequence conservation between human UbcH13 and the *I. scapularis* Bendless sequence, we hypothesized that the same would be true in ticks. We therefore employed a dual knock down scheme targeting both *uev1a* and *bendless* to assess their contribution in the *I. scapularis* IMD pathway during *A. phagocytophilum* infection.

4. Only Bendless-*uev1a* were silenced *in vivo* for *Anaplasma*. The same should be done with Relish and Caspar to prove that this is not an *in vitro* artifact.

In the revised manuscript additional experimental data has been included to address this point (Fig 4b and 4c). Significant knockdown was achieved with siRNA targeting either *relish* or *caspar*. In agreement with our previous *in vitro* findings with *A. phagocytophilum* and *in vivo* findings with *B. burgdorferi*, knocking down the positive regulator of the IMD pathway, *relish*, *in vivo* resulted in a significant increase in *A. phagocytophilum* colonization (2.6X more than ticks microinjected with the scrambled control; Figure 4b). Similarly, knocking down the negative regulator of the IMD pathway, *caspar*, *in vivo* caused a significant decrease in *A. phagocytophilum* colonization (6.3X less than ticks microinjected with the scrambled control).

5. Figure 5B. The values of dipteracin for the different concentrations of MPPC used in this experiment are higher than the dipteracin value for the lowest concentration of PODAG but only the latter is statistically significant. Please clarify.

We have now corrected this error (Fig. 5c). The lowest concentration of PODAG (0.01 ng) does not statistically upregulate *dipteracin* transcripts when evaluated by a one-tailed student's t-test (p value = 0.0755).

6. Figure 5c-d. Pre-incubation with 1ng of PODAG confers protection to ISE6 cells (ticks) but not to S2 cells (*Drosophila*). Could it be possible that PODAG elicit a higher response in tick cells than in *Drosophila* cells?

We thank the reviewer for pointing out this interesting possibility. Although there is a high degree of gene conservation between insects and arachnids, there is a significant amount of divergence observed when comparing the IMD pathway among arthropods (Palmer and Jiggins, 2015. *Mol Biol Evol*). Therefore, it is indeed possible, as the reviewer noted, that these two systems would not respond the same way when stimulated with some agonists.

Reviewer 3 (Tech. reviewer for lipidomics)

I have examined the manuscript from the point of view of a lipid expert only. The authors briefly state that the lipid composition of host free *A. phagocytophilum* was investigated by MALDI TOF/MS and that phosphatidylglycerol was the main lipid component. The experiments performed to test whether exogenous POPG, PODAG and MPPC could stimulate humoral immunity pathways look ok to me. However, a description of the methods of lipid analysis (lipid extraction method, matrix used in mass spectrometry analysis) is missing and should be added; if possible the lipid profile of the microorganism could be included in the manuscript (only as supplementary information of course) also because, to my knowledge, very little information on lipids of *A. phagocytophilum* is available in the literature.

We thank reviewer#3 for the thoughtful consideration of this study and apologize for not including these details in the original submission. We have now included detailed methods describing the lipid extraction and analysis (Pages 24-25, Lines 565-587) and a supplementary table outlining the overall output from this study (Supplementary Table 3).

REVIEWERS' COMMENTS:

Reviewer #1 (Remarks to the Author):

While all comments and concerns from the first round of review have been addressed, one concern remains. In particular, I remain concerned about drawing firm conclusions about gene functions (or lack thereof) when RNAi knockdown efficiency is only 30%. I realize this study is not in a typical model system, and that is clearly a strength of the study. However, the conclusion, and the language therein, should be adjusted so as to recognize that with only 30% knockdown, any negative conclusion, like the PGRPs are not involved, are soft.

Reviewer #3 (Remarks to the Author):

I have reconsidered the manuscript of Dr. Pedra and coworkers, focusing on the lipid analyses. I have carefully examined the corresponding new information in materials and methods section (rows 565-606) and the novel supplemental table 3 now in the manuscript. The only request from my side is to check whether reference 21 at row 567 is appropriate or not.

REVIEWERS' COMMENTS

Reviewer #1:

While all comments and concerns from the first round of review have been addressed, one concern remains. In particular, I remain concerned about drawing firm conclusions about gene functions (or lack thereof) when RNAi knockdown efficiency is only 30%. I realize this study is not in a typical model system, and that is clearly a strength of the study. However, the conclusion, and the language therein, should be adjusted so as to recognize that with only 30% knockdown, any negative conclusion, like the PGRPs are not involved, are soft.

We thank Reviewer 1 for this comment and agree that the language describing negative conclusions should be softened. We have now modified the language in the subsection titled “PGRPs do not play a role in the tick IMD pathway during *A. phagocytophilum* infection” to now read “PGRP knockdown does not impact *A. phagocytophilum* infection”. With this modification, we have removed any strong negative conclusions about the involvement of the soluble intracellular and/or extracellular PGRPs interfacing with the IMD pathway in ticks.

Reviewer #3:

I have reconsidered the manuscript of Dr. Pedra and coworkers, focusing on the lipid analyses. I have carefully examined the corresponding new information in materials and methods section (rows 565-606) and the novel supplemental table 3 now in the manuscript. The only request from my side is to check whether reference 21 at row 567 is appropriate or not.

We thank Reviewer 2 for bringing this to our attention and have now removed reference 21 from this section.